

# ProkEvo: an automated, reproducible, and scalable framework for high-throughput bacterial population genomics analyses

Natasha Pavlovikj[1,*], Joao Carlos Gomes-Neto[2,3,*], Jitender S. Deogun[1] and Andrew K. Benson[2,3]

[1] Department of Computer Science and Engineering, University of Nebraska-Lincoln, Lincoln, Nebraska, United States of America
[2] Department of Food Science and Technology, University of Nebraska-Lincoln, Lincoln, Nebraska, United States of America
[3] Nebraska Food for Health Center, University of Nebraska-Lincoln, Lincoln, Nebraska, United States of America
* These authors contributed equally to this work.

Corresponding authors
Natasha Pavlovikj,
natasha.pavlovikj@huskers.unl.edu
Andrew K. Benson,
abenson1@unl.edu

## ABSTRACT

Whole Genome Sequence (WGS) data from bacterial species is used for a variety of applications ranging from basic microbiological research, diagnostics, and epidemiological surveillance. The availability of WGS data from hundreds of thousands of individual isolates of individual microbial species poses a tremendous opportunity for discovery and hypothesis-generating research into ecology and evolution of these microorganisms. Flexibility, scalability, and user-friendliness of existing pipelines for population-scale inquiry, however, limit applications of systematic, population-scale approaches. Here, we present ProkEvo, an automated, scalable, reproducible, and open-source framework for bacterial population genomics analyses using WGS data. ProkEvo was specifically developed to achieve the following goals: (1) Automation and scaling of complex combinations of computational analyses for many thousands of bacterial genomes from inputs of raw Illumina paired-end sequence reads; (2) Use of workflow management systems (WMS) such as Pegasus WMS to ensure reproducibility, scalability, modularity, fault-tolerance, and robust file management throughout the process; (3) Use of high-performance and high-throughput computational platforms; (4) Generation of hierarchical-based population structure analysis based on combinations of multi-locus and Bayesian statistical approaches for classification for ecological and epidemiological inquiries; (5) Association of antimicrobial resistance (AMR) genes, putative virulence factors, and plasmids from curated databases with the hierarchically-related genotypic classifications; and (6) Production of pan-genome annotations and data compilation that can be utilized for downstream analysis such as identification of population-specific genomic signatures. The scalability of ProkEvo was measured with two datasets comprising significantly different numbers of input genomes (one with ~2,400 genomes, and the second with ~23,000 genomes). Depending on the dataset and the computational platform used, the running time of ProkEvo varied from ~3-26 days. ProkEvo can be used with virtually any bacterial species, and the Pegasus WMS uniquely facilitates addition or removal of programs from the workflow or modification of options within them. To demonstrate versatility of the ProkEvo platform, we performed a hierarchical-based population

structure analyses from available genomes of three distinct pathogenic bacterial species as individual case studies. The specific case studies illustrate how hierarchical analyses of population structures, genotype frequencies, and distribution of specific gene functions can be integrated into an analysis. Collectively, our study shows that ProkEvo presents a practical viable option for scalable, automated analyses of bacterial populations with direct applications for basic microbiology research, clinical microbiological diagnostics, and epidemiological surveillance.

## INTRODUCTION

Due to the advances in Whole Genome Sequence (WGS) technology, its decreasing costs, and the proliferation of publicly available tools and WGS-based datasets, the field of bacterial genomics is evolving rapidly from comparative analysis of a few representative strains of a given species, toward systematic, population-scale analyses of thousands of genomes. These large-scale analyses can provide new insights into evolutionary and ecological processes that alter the frequencies of different populations of pathogenic bacterial species in the environment and their transmission patterns to humans (*Quainoo et al., 2017*; *Pallen & Wren, 2007*; *Sheppard, Guttman & Fitzgerald, 2018*; *Land et al., 2015*). Applications of WGS-based population genomics range from basic research, public health, pathogen surveillance, clinical diagnostics, and ecological and evolutionary studies of pathogenic and non-pathogenic species (*Sheppard, Guttman & Fitzgerald, 2018*; *Joseph & Read, 2010*). Indeed, use of WGS by public health agencies is providing unprecedented levels of resolution and accuracy for source-tracking and WGS data is becoming the standard for epidemiological surveillance and outbreak detection (*Zhou et al., 2018*; *Alikhan et al., 2018*; *Dallman et al., 2015*).

While major applications of WGS-based genotyping in public health are focused on outbreak detection and source-tracking, the availability of large amounts of WGS data from populations of pathogenic bacteria from public health and regulatory agencies, and academic research creates tremendous opportunity for ecological and evolutionary inquiry at unprecedented scales of genomic resolution. For example, systematically monitoring the frequencies of specific variants of a pathogen, collected over time from the environment, food animals, and food production environments, can identify significant shifts in genotype frequencies that are driven by ecological events in the environment and/or within food production systems (*Yahara et al., 2017*). Powered statistically by the large number of genomes available from historical and ongoing surveillance, complex trait analyses can be used to identify causal variants and/or gene acquisition/loss events that are associated with changes in frequency of specific sub-populations (e.g., shifts from low-high frequency of isolation). Because these variants or gene acquisition/loss are likely to be causal, understanding their impact on gene function and pathways can illuminate adaptation and ecological fitness traits that influence survival in the environment and/or

transmission to humans (*Croucher et al., 2014*; *Yahara et al., 2017*). For example, candidate causal variants or gene acquisition/loss events associated with distinct populations at different scales of genotypic classification (e.g., serovars (*Ingle et al., 2016*; *Yoshida et al., 2016*), or sub-populations (*Sheppard, Jolley & Maiden, 2012*)), can further be examined in silico to predict unique functional characteristics and phenotypes of populations (e.g., antimicrobial resistance (AMR)) (*McDermott et al., 2016*), virulence, and metabolic attributes (*Yahara et al., 2017*; *Laabei et al., 2014*). Such predictions can further be incorporated into hypothesis-testing empirical measurements of predicted phenotypes in vitro.

To understand the relationship of genomic variation, evolutionary and ecological processes, it is necessary to classify isolates of a given species genotypically at various levels of resolution. WGS data provides the basis for such classifications and currently, there are small number of automated pipelines available for analysis and genotypic classification of bacterial genomes, including EnteroBase (*Zhou et al., 2019*), TORMES (*Quijada et al., 2019*), Nullarbor (*Seemann et al., 2020*), ASA$^3$P (*Schwengers et al., 2020*), Bactopia (*Petit & Read, 2020*). These pipelines each have unique characteristics and were developed for different purposes. They also differ in the programming language used, the size and type of supported input data, the supported bioinformatics tools, and the computational platform used. The pipelines do share some elements of genotypic classification at different levels of resolution, but the classifications and scalability vary. Our work was motivated by the need for a reproducible, automated, flexible, and portable, WGS-based population genomics platform that can accommodate scalable, hierarchical-based genotypic classifications and gene annotations for high-throughput, population-based inquiry. To accommodate the complex combinations of multiple, sequential data processing steps required for such a platform, which inevitably demands an amalgamation of various software, we used a highly optimized Workflow Management System (WMS) (*Koster & Rahmann, 2012*; *Di Tommaso et al., 2017*; *Apache Airflow, 2015*; *Deelman et al., 2005*), that can efficiently manage massive numbers of computational operations in different types of high-performance computing environments, including University or publicly available clusters (*HCC, 2008*; *Towns et al., 2014*), clouds (*Langmead & Nellore, 2018*), or distributed grids (*Pordes et al., 2007*; *Sfiligoi et al., 2009*).

In this paper, we describe ProkEvo–an automated and user-friendly platform for population-based inquiry of bacterial species that is designed to provide hierarchical-based genotypic classifications and association of accessory genomic content (e.g., AMR, virulence genes, pan-genomic content) in a scalable platform. The ProkEvo platform manages the large number of bioinformatics programs and their dependencies through the Pegasus WMS and is portable to computing clusters, clouds, and distributed grids. ProkEvo works with raw paired-end Illumina reads as input, and is composed of multiple sequential steps for processing and analysis of data that is scalable from hundreds to many thousands of genomes. For each input genome, these steps include trimming and quality control, genome assembly, serovar prediction (in the case of *Salmonella enterica*), hierarchical-based genotypic classification based on legacy multilocus-sequence typing (MLST) using seven loci or core-genome MLST (cgMLST) using approximately 300 loci,

and hierarchical variant classification based on Bayesian nested clustering analysis at different scales of resolution. ProkEvo also associates content of AMR genes, putative virulence genes, plasmids, and pan-genomic content with the hierarchical-based genotypic classifications.

Here, we show the utility and adaptability of ProkEvo for basic metrics of population genomics analysis on three different bacterial pathogens (*Salmonella enterica, Campylobacter jejuni* and *Staphylococcus aureus*). We also demonstrate the scalability and modularity of ProkEvo with datasets ranging from ~2,400 to ~23,000 genomes and further illustrate the portability and performance of ProkEvo on two different computational platforms, the University of Nebraska high-performance computing cluster (Crane) and the Open Science Grid (OSG), a distributed, high-throughput computational platform. Because of the multi-disciplinary environments required for implementation and applications of ProkEvo, we also provide guidance for researchers on utilization of some of the output files generated by ProkEvo to perform meaningful hierarchical-based population analyses in a reproducible fashion using a combination of R and Python scripts.

## MATERIALS & METHODS

### Overview of ProkEvo

The ProkEvo pipeline is capable of processing raw, paired-end Illumina reads obtained from tens of thousands of genomes present in the NCBI database utilizing high-performance and high-throughput computational resources. The pipeline is composed of two sub-pipelines: (1) The first sub-pipeline performs the standard data processing steps of sequence trimming, de novo assembly, and quality control; (2) The second sub-pipeline uses the assemblies that have passed the quality control and performs specific population-based classifications (serotype prediction specifically for Salmonella, genotype classification at different scales of resolution, analysis of core- and pan-genomic content). Pegasus WMS manages and splits each sub-workflow into as many independent tasks as possible to take advantage of many computational resources.

A text file of SRA identifications corresponding to raw Illumina reads available from the Sequence Read Archive (SRA) database in NCBI (NCBI SRA) is used as an input to the pipeline. The first step of the pipeline and the first sub-workflow is automated download of genome data from NCBI SRA (*Leinonen, Sugawara & Shumway, 2010*). This is done using the package parallel-fastq-dump (*Valieris, 2020*). The SRA files are downloaded using the prefetch utility, and the downloaded files are converted into paired-end fastq reads using the program parallel-fastq-dump. While the SRA Toolkit (*Leinonen, Sugawara & Shumway, 2010*) provides the same functionality, this toolkit can be slow sometimes and show intermittent timeout errors, especially when downloading many files. parallel-fastq-dump is a wrapper for SRA Toolkit that speeds the process by dividing the conversion to fastq files into multiple threads. Beside downloading raw Illumina reads from NCBI, ProkEvo also supports usage of already locally downloaded fastq reads. In this case, the absolute path to the reads should be specified in the replica catalog provided by Pegasus. More details about this setup are available on the documentation page of

ProkEvo (https://github.com/npavlovikj/ProkEvo/wiki/3.1.-Setup-on-high-performance-computing-cluster#2-using-already-downloaded-raw-reads). After the raw paired-end fastq files are generated, quality trimming and adapter clipping is performed using Trimmomatic (*Bolger, Lohse & Usadel, 2014*). FastQC is used to check and verify the quality of the trimmed reads (*Andrews, 2010*) and it is run independently for each paired-end dataset with concatenation of all output files at the end for a summary. The paired-end reads are assembled de novo into contigs using SPAdes (*Bankevich et al., 2012*). These assemblies are generated using the default parameters. The quality of the assemblies is evaluated using QUAST (*Gurevich et al., 2013*). The information obtained from QUAST is used to discard assemblies with 0 or more than 300 contigs, or assemblies with N50 value of less than 25,000. These cutoff values vary between species, and if needed, they can be modified by the researcher before running ProkEvo. Examples of how this can be done are provided in the documentation page of ProkEvo (https://github.com/npavlovikj/ProkEvo/wiki/4.3.-Change-running-options-for-existing-tool-in-ProkEvo). All the modifications should be done before running ProkEvo. QUAST-based filtering of the assemblies concludes the first part or first sub-pipeline of the workflow. Each of these steps is independent of the input data and each task is performed on one set of paired-end reads using one computing core. This makes the analyses modular and suitable for high-throughput resources with many available cores. Moreover, having many independent tasks significantly reduces the memory and time requirements while generating the same results as when the analyses are done sequentially. Thus, if a dataset has paired-end reads from $n$ different genomes and a computational platform has $n$ available cores (1:1 correspondence), ProkEvo will scale and utilize all these resources at the same time.

The second sub-pipeline uses the assemblies which passed quality control to perform specific population-based characterizations, including genotypic classifications, serovar prediction (exclusively for Salmonella), gene-based annotations, and pan-genome outputs. PlasmidFinder is used to identify plasmids in the assemblies (*Carattoli et al., 2014*). PlasmidFinder comes with curated database of plasmid replicons to identify plasmids in the WGS data (currently over-represented plasmids from the Enterobacteriaceae). SISTR is used for Salmonella and produces serovar prediction and in silico molecular typing by determination of core-genome multilocus-sequence typing (cgMLST) gene alleles (~330 loci) (*Yoshida et al., 2016*). SISTR generates multiple output files. Of primary interest for downstream analyses is the main SISTR output file named sistr_output.csv. The filtered assemblies are annotated using Prokka (*Seemann, 2014*), which is based on a curated set of core and HMM databases for the most common bacterial species. If needed, one can customize and create their own annotation database. In addition to the other files, Prokka produces annotation files in GFF3 format that are used with Roary (*Page et al., 2015*) to identify the pan-genome and to generate core-genome alignments. The core-genome alignment file produced is then used with fastbaps, an improved version of the BAPS clustering method (*Tonkin-Hill et al., 2019*), to hierarchically cluster the genomic sequences from the multiple sequence alignment in varying numbers of stratum (i.e., levels of resolution). Multilocus-sequence typing is also performed on the assemblies

using MLST (*Seemann, 2020a*). Here, the filtered genome assemblies from individual bacterial isolates are categorized into specific variants based on allele combinations from seven ubiquitous, house-keeping genes (*Jolley & Maiden, 2010*). In addition to these analyses, the filtered assemblies are screened for AMR and virulence associated loci using ABRicate (*Seemann, 2020b*). ABRicate comes with multiple comprehensive gene-based mapping databases, and the ones used in ProkEvo are NCBI (*Feldgarden et al., 2019*), CARD (*Jia et al., 2016*), ARG_ANNOT (*Gupta et al., 2013*), Resfinder (*Zankari et al., 2012*), and VFDB (*Chen et al., 2015*). Prokka, SISTR, PlasmidFinder, MLST, and ABRicate are independent of each other, and they are all run simultaneously in parallel. Moreover, Prokka, SISTR, and PlasmidFinder perform their computations per filtered assembly, while MLST and ABRicate use all filtered assemblies together. Running multiple independent jobs simultaneously is one of the key factors to maximize computational efficiency. With respect to Salmonella genomes, once the SISTR analyses finish for all assemblies, the generated independent sistr_output.csv files are concatenated. This aggregation of files can be done because the genome categorization to serovars and cgMLST variants done by SISTR occurs completely independent for each genome. Each tool executed in ProkEvo is run with specific options set as defaults. While the options used in this paper fit the presented case studies, these options are easily adjustable and configurable in the pipeline. Because we developed ProkEvo for studying a diverse array of bacterial species, the pipeline was specifically designed to incorporate programs such as SISTR for *Salmonella enterica*, where serovar classifications can be made accurately based on the Kauffman-White scheme (*Rowe & Hall, 1989*). However, other serotype prediction modules can be substituted for SISTR to accommodate user-specific needs. Additionally, the MLST program can be directed to species-specific sets of genetic loci used for classification, as shown with the *Campylobacter jejuni* and *Staphylococcus aureus* datasets.

The modularity of ProkEvo allows us to decompose the analyses into multiple tasks, some of which can be run in parallel, and utilize a WMS. ProkEvo is dependent on many well-developed bioinformatics tools and databases. A list of the exact versions of the bioinformatics tools and databases used for reproducing the analyses in this paper is given on Table S1. The setup and the installation of the needed tools, dependencies and databases are not always trivial. To make this process easier, reduce the technical complexity, and allow reproducibility, we provide two software distributions for ProkEvo. The first distribution is a conda environment based on a yaml file that contains all software dependencies and versions utilized (*Anaconda, 2012*), and the second one is a Docker image that can be used with Singularity (*Docker, 2013*). Both distributions are supported by the majority of computational platforms and integrate well with ProkEvo, and can be easily modified to include other tools and steps. The software dependencies in the conda yaml file and Docker image are pinned to their specific versions used for the analyses in this paper in order to provide reproducibility. By default, when run, ProkEvo creates conda environment with all needed tools and databases, so the researcher does not need to do any separate setup for the dependencies. The code for ProkEvo, and both the conda yaml file and the Docker image, are publicly available at our GitHub repository

(https://github.com/npavlovikj/ProkEvo) and (https://github.com/npavlovikj/ProkEvo/tree/master/distribution) respectively.

## Features of ProkEvo

The provided distribution of ProkEvo is generalized to work for multiple population-based applications and generate results with minimal effort for implementation. Because researchers may need to modify, optimize, or expand ProkEvo for their own needs, we have designed ProkEvo with capabilities for easy customization through the Pegasus WMS. The default commands and settings of the bioinformatics tools used in ProkEvo are documented on our documentation page (https://github.com/npavlovikj/ProkEvo/wiki/2.1.-Bioinformatics-tools-and-commands-used). These settings or modifications to them should be applied before submitting and running ProkEvo. ProkEvo supports various advanced features, such as:

1. Adding new bioinformatics tool to ProkEvo (https://github.com/npavlovikj/ProkEvo/wiki/4.1.-Add-new-bioinformatics-tool-to-ProkEvo)
2. Removing bioinformatics tool from ProkEvo (https://github.com/npavlovikj/ProkEvo/wiki/4.2.-Remove-existing-bioinformatics-tool-from-ProkEvo)
3. Changing options for already existing tool in ProkEvo (https://github.com/npavlovikj/ProkEvo/wiki/4.3.-Change-running-options-for-existing-tool-in-ProkEvo)
4. Running ProkEvo on Virtual Cloud Machine (https://github.com/npavlovikj/ProkEvo/wiki/3.2.-Setup-on-virtual-cloud-machine)

## Pegasus workflow management system

ProkEvo uses the Pegasus WMS, which is a framework that automatically translates abstract, high-level workflow descriptions into concrete efficient scientific workflows that can be executed on different computational platforms such as clusters, grids, and clouds. The abstract workflow of Pegasus WMS contains information and description of all executable files (transformation catalog) and logical names of the input files used by the workflow (replica catalog). Complementing the abstract component is a concrete workflow, which specifies the location of the data and the execution platform (*Deelman et al., 2005*). The workflow is organized as a directed acyclic graph (DAG), where the nodes are the tasks and the edges are the dependencies. Next, the workflow is submitted using HTCondor (*HTCondor, 1988*). Pegasus WMS uses DAX (directed acyclic graph in XML) files to describe an abstract workflow. These files can be generated using programming languages such as Java, Perl, or Python. The high-level of abstraction of Pegasus allows users to ignore low-level configurations required by the underlying execution platforms. Pegasus WMS is an advanced system that supports data management and task execution in automated, reliable, efficient, and scalable manner. This whole process is monitored, and the workflow data is tracked and staged. The requested output results are presented to the users, while all intermediate data can be removed or re-used. In case of errors, jobs are automatically re-initiated. If the errors persist, a checkpoint file is produced so the job can be resubmitted and resumed. Pegasus WMS supports sub-

workflows, task clustering and defining memory and time resources per task. Pegasus WMS also generates web dashboard for each workflow for better workflow monitoring, debugging, and analyzing, which helps users to analyze workflows based on useful statistics and metrics of the workflow performance, running time, and machines used.

ProkEvo uses Python to create the workflow description. Each step of the pipeline is a computational job represented as a node in the DAG. Two nodes are connected with an edge if the two jobs need to be run one after another. The input and output files are defined in the DAG as well. All jobs that are not dependent on each other can be run concurrently. Each job uses its own predefined script that executes the program required by the job with the specified options. This script can be written in any programming language. The specific versions of the bioinformatics tools and programs required by ProkEvo can be distributed through conda environment with provided yaml file (https://github.com/npavlovikj/ProkEvo/blob/master/distribution/prokevo.yml) or Docker image (https://github.com/npavlovikj/ProkEvo/blob/master/distribution/Dockerfile). The predefined scripts within this release of ProkEvo enable running without further change or modification. With the modularity of Pegasus, each job requests its own run time and memory resources. Exceeding the memory resources is a common occurrence in any bioinformatics analysis and based on this assumption, when exceeding the memory is a reason for a job failure, Pegasus retries the job with increased requirements. Higher memory requirements may imply longer waiting times for resources, and the Pegasus WMS uses high memory resources only when needed. ProkEvo is developed in a way that supports execution on various high-performance and high-throughput computational platforms. In the analyses for this paper, we use both the University cluster and OSG, and working versions for both platforms are available in our GitHub repository (https://github.com/npavlovikj/ProkEvo).

## Computational execution platforms

Traditionally, data-intensive scientific workflows have been executed on high-performance and high-throughput computational platforms. While high-performance platforms provide resources for analyses that require significant numbers of cores, time, and memory, high-throughput platforms are suitable for many small and short independent tasks. The design of ProkEvo is suitable for different computational environments like University and other publicly or privately available clusters and grids, and thus provides flexibility in the computational platform. We have evaluated ProkEvo on two different computational platforms—a University cluster and the distributed Open Science Grid.

## University cluster (Crane), a high-performance computational platform

University and other public clusters are shared by diverse communities of users and enforce fair-share scheduling and file and disk spaces quotas. These clusters are suitable for various types of jobs, such as serial, parallel, GPU, and high memory specific jobs, thus the high-performance. Crane (*HCC, 2008*) is one of the high-performance computing clusters at the University of Nebraska Holland Computing Center (HCC). Crane is Linux cluster, having 548 Intel Xeon nodes with RAM ranging from 64 GB to 1.5 TB, and it

supports Slurm and HTCondor as job schedulers. In order to use Crane, users obtain an HCC account associated with a University of Nebraska faculty or research group. Importantly, most University and publicly available high-performance clusters are administered in a manner similar to Crane and would be suitable for running ProkEvo.

Crane has support for Pegasus and HTCondor, and no further installation is needed in order to run ProkEvo. Due to the limited resources and fair-share policy on Crane, tens to hundreds of independent jobs can be run concurrently. We provide a version of ProkEvo suitable for Crane with conda yaml file, which contains all required software and its specific versions used in this paper (https://github.com/npavlovikj/ProkEvo/blob/master/distribution/prokevo.yml). Crane has a shared file system where the data is accessible across all computing nodes. Depending on the supported file system, Pegasus is configured separately and handles the data staging and transfer accordingly. However, users do not need advanced experience in high-performance computing to run ProkEvo on Crane, or most other University or publicly available clusters. Users only need to provide list of SRA identifications and run the submit script that distributes the jobs automatically as given in our GitHub repository (https://github.com/npavlovikj/ProkEvo/wiki/3.1.-Setup-on-high-performance-computing-cluster).

## Open Science Grid (OSG), a distributed, high-throughput computational platform

The Open Science Grid (OSG) is a distributed, high-throughput computational platform for large-scale scientific research (*Pordes et al., 2007*; *Sfiligoi et al., 2009*). OSG is a national consortium of more than 100 academic institutions and laboratories that provide storage and tens of thousands of resources to OSG users. These sites share their idle resources via OSG for opportunistic usage. Because of its opportunistic approach, OSG as a platform is ideal for running massive numbers of independent jobs that require less than 10 GB of RAM, less than 10 GB of storage, and less than 24 h running time. If these conditions are fulfilled, in general, OSG can provide unlimited resources with the possibility of having hundreds or even tens of thousands of jobs running at the same time. The OSG resources are Linux-based, and due to the different sites involved, the hardware specifications of the resources are different and vary. Access and use of OSG is free for academic purposes and the user's institution does not need to be part of OSG to use this platform.

All steps from the population genomics analyses of ProkEvo fulfill the conditions for OSG-friendly jobs and ProkEvo can efficiently utilize these distributed high-throughput resources to run thousands of analyses concurrently when the resources are available. OSG supports Pegasus and HTCondor, so no installation steps are required. We provide version of ProkEvo suitable for OSG (https://github.com/npavlovikj/ProkEvo/tree/master/OSG). This version uses the Docker image with all specific releases of the software requirements via Singularity and supports non-shared file system (https://github.com/npavlovikj/ProkEvo/blob/master/distribution/Dockerfile). In non-shared systems, the resources do not share the data. The data are read and written from a staging location, all of which is managed by the Pegasus WMS. In order to run ProkEvo on OSG, users only

need to provide list of SRA identifications and run the submit script without any advanced experience in high-throughput computing.

## Population genomics analyses

The population-based analyses performed in this paper provide an initial guidance on how to comprehensively utilize the following output files produced by ProkEvo for hierarchical-based genotypic classifications. These classifications are based on: (1) MLST output (.csv) (*Seemann, 2020a*); (2) SISTR output (.csv) (*Yoshida et al., 2016*); (3) BAPS output (.csv) (*Tonkin-Hill et al., 2019*); (4) Roary generated core-genome alignment (core_gene_alignment.aln) and accessory-genome (accessory_binary_genes.fa.newick) files for both phylogenetic and dendogram/clustering analysis, respectively (*Page et al., 2015*); and 5) ABRicate output (.csv) containing AMR genes using the Resfinder database (*Zankari et al., 2012*). We use both R (version 4.0.3) and Python 3 (version 3.8.3) Jupyter Notebooks (version 6.0.3) for all our initial guidance into combining some of the outputs for population-based data analyses (https://github.com/npavlovikj/ProkEvo/tree/master/jupyter_r_notebooks). The specific Python libraries used were pandas (version 1.0.2), numpy (version 1.18.1), matplotlib (version 3.2.1), seaborn (version 0.10.1). The specific R libraries used were tidyverse (version 1.3.0), ggplot2 (version 3.3.2), ggtree (version 2.2.4). The input data used for these analyses is available on Figshare (https://figshare.com/projects/ProkEvo/78612).

A first general step in this type of analysis is opening all files in the preferred environment (i.e., RStudio or JupyterHub), and merging them into a single data frame based on the SRA (genome) identification. Next, we perform quality control (QC) of the data, focusing on identifying and dealing with missing values, or cells of the data frame containing erroneous characters such as hyphens (-) and interrogation marks (?). For that, we demonstrate our approach for cleaning up the data prior to conducting exploratory statistical analysis and generating all visualizations.

In the case of Salmonella datasets, an additional "checking/filtering" step was used after the QC is complete. Since the program SISTR provides a serovar call based on genotypic information, one can opt for keeping or excluding those genomes that do not match the original serovar identification in the analysis. Both approaches are justifiable with the latter one being more conservative, and it specifically assumes that the discordance between data entered in NCBI and genotypic prediction done by SISTR is accurate. However, it is important to remember that we initially expect that the dataset belongs to a particular serovar because of the keywords we used to search the NCBI SRA database, such as: "*Salmonella* Newport", "*Salmonella* Typhimurium", or "*Salmonella* Infantis". Typically, the proportion of genomes that are classified differently by SISTR than the designation associated with the file in SRA is ~<3% for any given Salmonella dataset tested here. In our application, we chose a conservative approach and either filtered the "miscalls" out of the data, or kept it as a separate group called "other serovars". The latter approach was done for specific analyses, such as phylogenetics, where the program of choice used for data visualization requires all data points to be in place (e.g., ggtree version 2.2.4 in R version 4.0.3). This situation arises because the core-genome alignment used for

the phylogeny is generated by Roary without considering the SISTR prediction for serovar calls. If such consideration is relevant, the user can add a condition to the pipeline to run Roary after considering SISTR results, but this situation only applies to Salmonella genomes. However, we do note that stringent requirements for serotype classification (i.e., filtering out "miscalls" based on SISTR predictions) could eliminate important variants that may genotypically match known populations of the serovar, but which have acquired mutations or recombination events at serotype-determining loci. Our suggestion is that for any predictive analysis, one should either filter out, or classify the potential miscalls, or at least measure its contribution and effects on data interpretation after running SISTR.

To define hierarchical relationships of genotypic classifications at varying levels of resolution, the ProkEvo pipeline combines multi-locus MLST-based variants at different scales of resolution with Bayesian-based nested clustering analysis (BAPS), which classifies genomes based on core-genomic structure (i.e., only shared content). The BAPS-based approach to genomic classifications is callable, and allows the user to circumvent computationally-intensive use of phylogeny, which is not scalable to thousands of core genomes. Thus, evolutionary "familial" relationships across STs or thousands of cgMLST variants can be inferred by their hierarchical relationships to BAPS-based classifications. In this version of ProkEvo, we have implemented legacy MLST for ST calls using seven loci, cgMLST that uses approximately 330 loci for MLST analysis in the case of Salmonella, and a BAPS haplotype/sub-group classification using six layers of BAPS (BAPS1 being the lowest level of resolution and BAPS6 being the highest—top-down stratification). That is, BAPS1 represents the first level of resolution, within which sub-groups or multiple haplotypes will be formed (nested approach). The more levels of resolution used, the higher the degree of granularity (more sub-groups will be formed within BAPS levels) achieved while stratifying a population. To explore the hierarchical relationships of variants, one can simply examine the distribution of legacy STs among genomes belonging to identical or distinct clusters based on classification at the lowest level of BAPS resolution (BAPS1). Likewise, the genetic relationships of thousands of cgMLST variants can also be assessed with respect to the BAPS-based and ST-linked genomic architecture at different levels of BAPS resolution to infer evolutionary familial relationships. For instance, a highly clonal population of a single cgMLST variant would be expected to group into a single BAPS sub-group/haplotype at the lowest level of BAPS1 resolution, and remain confined to one or a small number of BAPS sub-groups at increasing levels of BAPS-based resolution (i.e., BAPS5-BAPS6). In contrast, a diverse population of cgMLST variants that are more distantly related (e.g., not highly clonally related) will partition between multiple BAPS sub-groups at higher levels of resolution, say BAPS5 or BAPS6. In practice, this analysis is important for examining the degrees of population heterogeneity and diversification, which has implications for ecological and epidemiological inference.

The above mentioned hierarchical approach was possible for the *S*. Newport dataset of ~2,400 genomes (USA data), but the core-genome alignment step, generated by Roary under our specific settings, was not scalable to the 10-fold larger dataset of *S*.

Typhimurium (~23,000 genomes—worldwide data). This larger dataset was split into twenty smaller datasets during the core-genome alignment step. Although random partitioning of the subsets should yield the same classifications of dominant genomic groups, the BAPS clustering will not necessarily assign the same genomic types in different datasets to the same sub-group/haplotype numbers. Thus, aggregation of the BAPS data from multiple, independently analyzed subsets requires user-based input. On the other hand, sub-setting larger datasets is advantageous for downstream data science and machine learning analyses, since they require a nested cross-validation approach for feature selection and predictive analytics. Herein, we used a random sampling approach to create subsets of the genomic data for the large number of S. Typhimurium genomes that were input into Roary. Based on the number of genomes, we created 20 subsets, each having 1,076-1,077 genomes. Obviously, downsampling is also possible provided one has a priori definition of the population structure, and/or other sources of information such as epidemiological data from outbreaks. Next, from the GFF files produced by Prokka, we randomly selected and assigned genomes to each group using custom Bash scripts. Both Roary and fastbaps were run per group, resulting in 20 independent runs with the corresponding output files. To evaluate randomness of subset assignments, the distribution of the major ST and cgMLST variants were assessed (https://github.com/npavlovikj/ProkEvo/blob/master/jupyter_r_notebooks/salmonella_typhimurium_analysis.ipynb). Subsequently, the population of S. Typhimurium was analyzed using its hierarchical structure simply going from ST to cgMLST variants.

Complementary to this population structure analysis, we also measured distributions of AMR genes within and between Salmonella serovars, including S. Infantis (~1,700 genomes—USA data). Within serovar, the relative frequencies of AMR genes were estimated between major ST variants using the ABRicate outputs from the Resfinder database for identification of putative AMR genes. We arbitrarily selected genes with proportion higher than or equal to 25% for S. Newport, S. Infantis, and S. Typhimurium, for visualizations, which were produced with ggplot2 in R (*Wickham, 2011*). The respective scripts are provided in our repository (https://github.com/npavlovikj/ProkEvo/blob/master/jupyter_r_notebooks/salmonella_abx.Rmd).

To demonstrate the versatility of ProkEvo across multiple species, we also conducted a population-based analysis of *C. jejuni* and *S. aureus* datasets comprising isolates from the USA, containing 21,919 and 11,990 genomes, respectively. For both datasets, we analyzed the population structure using BAPS1 and STs. The same hierarchical population basis described for Salmonella applies here, with BAPS1 coming first and STs next in terms of population ranking. We used a random sample of ~1,000 genomes of each species to demonstrate the distribution of BAPS1 and STs onto the phylogenetic structure. Phylogenies were constructed using the core-genome alignment produced by Roary, and by applying the FastTree program (*Price, Dehal & Arkin, 2010*) using the generalized time-reversible (GTR) model of nucleotide evolution without removing genomic regions putatively affected by recombination (https://github.com/npavlovikj/ProkEvo/blob/master/jupyter_r_notebooks/campylobactera_jejuni_s_aureus.Rmd). Additionally, we showed the distribution of STs within each bacterial species (only showed STs with

proportion higher than 1%), and the relationship between the relative frequencies of dominant STs and AMR genes. Genes with relative frequency below 25% were filtered out of the data. All visualizations were generated with ggplot2 (version 3.3.2) in R (version 4.0.3), and the scripts are also provided in our repository. All procedures used from quality control of the data all the way to tabular formatting and filtering were done with base R and tidyverse (version 1.3.0).

Lastly, we compared two options for integrative phylogenetic and population structure visualization using two software packages: ggtree version 2.2.4 using R version 4.0.3 vs. an online platform named phandango version 1.3.0 (https://jameshadfield.github.io/phandango/#/) (*Hadfield et al., 2018*). This analysis was done using both *C. jejuni* and *S. aureus* datasets. For *C. jejuni* and *S. aureus*, we randomly (random sampling without replacement) selected 1,044 and 1,193 genomes from the total population collected from NCBI comprised of 21,919 and 11,990 genomes, respectively—all of which were initially processed through ProkEvo, as described above. Phylogenetic trees were constructed for both samples using FastTree. Hierarchical population structure analysis was done using two layers of genotypic information: BAPS1 and ST classifications. Considering the potential impact of sample size in phylogenetic visualization due to varying branch length, we also generated phylogenies using randomly selected datasets of increasing size: 180, 360, 540, and 720 genomes for *C. jejuni*; and using 140, 350, 560, and 770 genomes for *S. aureus*. These genomes, belonging to varying subsets of different sample sizes, were selected upon classifying 18,845 of the 21,919 genomes of *C. jejuni*, and 11,597 of the 11,990 genomes of *S. aureus*, respectively, using the MLST approach with seven loci. Of note, all sampling was done randomly without replacement, and evenly across major STs (arbitrary cutoffs of 3% and 1% for including STs into major groups based on relative frequencies for either *C. jejuni* or *S. aureus*, respectively); whereas, the remainder ones were aggregated as "Other STs". Herein, our strategy was to first examine the population structure of each species using the ST genotyping to sample evenly across the most dominant STs, in order to avoid bias while constructing the random data subsets. Our choice to use ST instead of BAPS1 for a prior population structure assessment was due to the first being stable across runs, while the latter may have varying sub-group membership due to the randomness of its algorithm. This approach allowed us to specifically test what impact the plotting program would have on visualizing the phylogenetic topology and branching patterns. Of note, both programs required the phylogenetic tree (.tree) and metadata (population structure—.csv) files as input.

# RESULTS

## Overview of ProkEvo

In Table 1, we provide a comparative analysis depicting some of the main similarities and differences between ProkEvo and major pipelines that are publicly available for comparative and population-based bacterial genomics analyses. The overall flow of tasks performed in ProkEvo is illustrated in Fig. 1, including all specific bioinformatics tools used for each task. A list of the exact versions of all the tools and databases used in ProkEvo is shown on Table S1. The DAG shown in Fig. 2 represents the Pegasus WMS design of

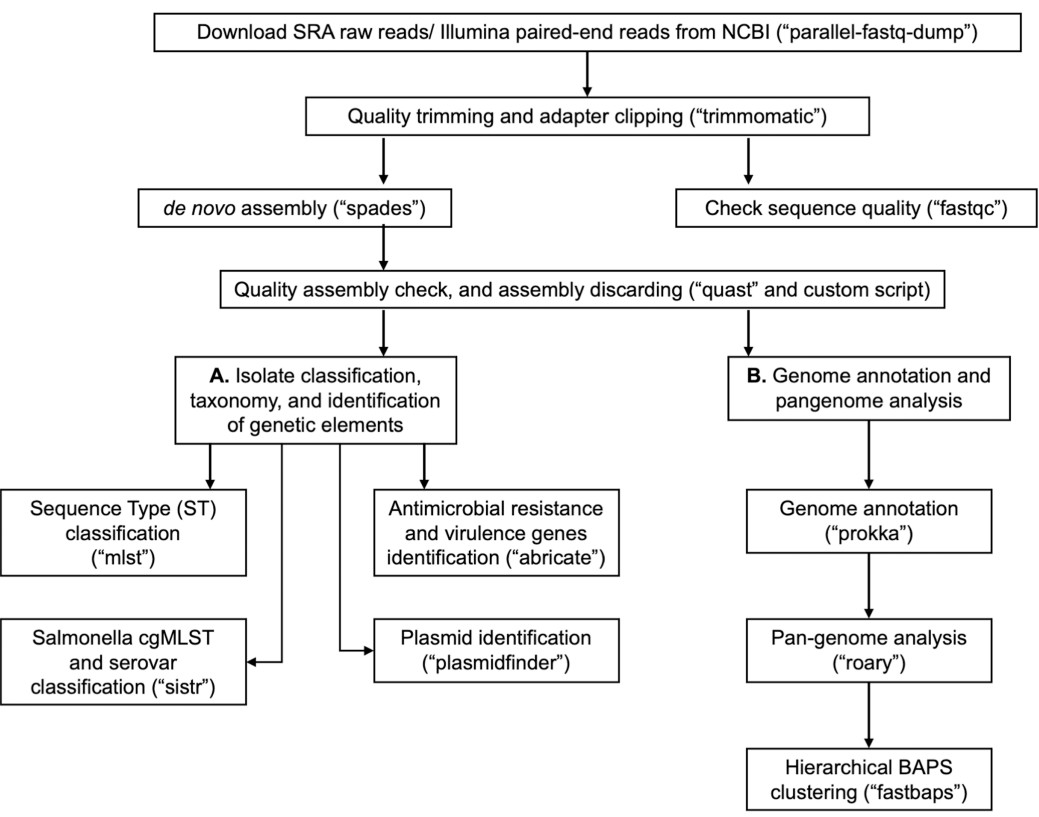

**Figure 1 Overall ProkEvo's computational workflow.** Top-down flow of tasks for the ProkEvo pipeline. The squares represent the steps, where the bioinformatics tool used for each step is shown in brackets. The pipeline starts with downloading raw Illumina sequences from NCBI, after providing a list of SRA identifications, and subsequently performing quality control. Next, de novo assembly is performed on each genome using SPAdes and the low-quality contigs are removed. This concludes the first part of the pipeline, the first sub-workflow. The second sub-workflow is composed of more specific population-genomics analyses, such as genome annotation and pan-genome analyses (with Prokka and Roary), isolate cgMLST classification and serotype predictions from genotypes in the case of Salmonella (SISTR), ST classification using the MLST scheme, non-supervised heuristic Bayesian genotyping approach using core-genome alignment (fastbaps), and identifications of genetic elements with ABRicate and PlasmidFinder.

ProkEvo and it shows all independent input and output files, tasks, and the dependencies among them. The modularity of ProkEvo allows every single task to be executed independently on a single core. As seen on Fig. 2, there are approximately 10 tasks executed per genome. When ProkEvo is used with whole bacterial populations of thousands of genomes, the number of total tasks is immense.

To evaluate the capability of the Pegasus WMS to scale tasks independently on diverse computational platforms, ProkEvo was run with two datasets of significantly different size (~2,400 genomes [1X] vs. ~23,000 genomes [10X]) on two different computational platforms—the University of Nebraska high-performance computing cluster (Crane) and the Open Science Grid (OSG), a distributed, high-throughput cluster (Fig. 3). The ProkEvo code available on our GitHub page supports both platforms and each platform has unique structure and idiosyncratic advantages and disadvantages (Fig. 3). Each dataset was

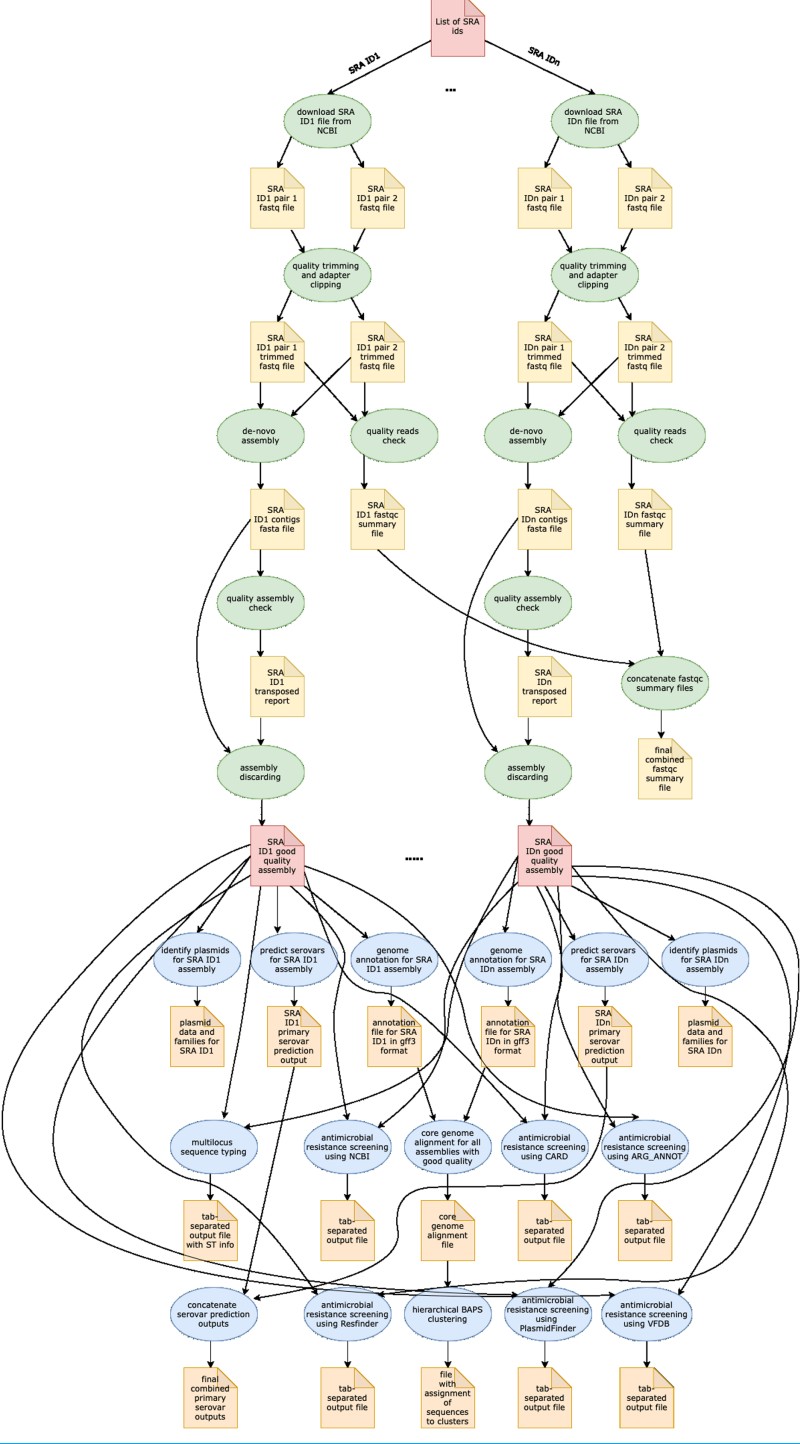

**Figure 2 Pegasus workflow of ProkEvo.** Pentagons represent the input and output files, the ovals represent the tasks (jobs), and the arrows represent the dependency order among tasks. Pentagons are colored in red for the input files used for the first and second sub-workflow, respectively. The yellow pentagons and the green ovals represent the input and output files, and tasks (jobs) that are part of the first sub-workflow. The pentagons colored in orange and the ovals colored in blue are the input and output files, and tasks used in the second sub-workflow. While the first sub-workflow is more modular, most of the tasks from the second sub-workflow are performed on all processed genomes together. Here, the steps of the analyses for two genomes are shown, and those steps and tasks remain the same regardless

**Figure 2** (continued)
of the number of genomes. The number of tasks significantly increases with the number of genomes used, and because of the modularity of ProkEvo, each task is run on a single core which facilitates parallelization at large scale. Theoretically, if there are $n$ cores available on the computational platform, ProkEvo can utilize all of them and run the corresponding $n$ independent tasks, simultaneously (1:1 correspondence).

run once on the two platforms and performance metrics were collected for the Pegasus WMS workflow. Of note, there may be variation in the ProkEvo runtime from project to project based on the availability of resources on each platform. As an HPC resource of the Holland Computing Center, the Crane cluster is managed by fair-share scheduling, while as an opportunistic HTC resource, the OSG resources may be dynamically de-provisioned or having intermittent issues. These factors may impact the future predictability of running time and performance of ProkEvo on both platforms. On average, we had hundreds of jobs running at a time on Crane, and because of the similar type of nodes available, the runtime should be similar for multiple runs of the same workflow. On the other hand, the nodes on OSG are more diverse and the runtime and the number of jobs for multiple runs can be significantly different (from few jobs running at the same time to few tens of thousand).

ProkEvo consists of two sub-workflows, with number of jobs varying from a few thousands to a few hundreds of thousands, depending on the dataset. "pegasus-statistics" generates summary metrics/statistics regarding the workflow performance, such as the total number of jobs, total run time, number of jobs that failed and succeeded, task and facility information, etc. The total distributed running time is the total running time of ProkEvo from the start of the workflow to its completion. The total sequential running time is the total running time if all steps in ProkEvo are executed one after another. In case of retries, the running times of all re-attempted jobs are included in these statistics as well. Beside the workflow runtime information, Table 2 also shows the maximum total number of independent jobs ran on Crane and OSG within one day. Moreover, the total count of succeeded jobs is shown for both computational platforms and datasets.

When run on Crane, ProkEvo with *S.* Newport (~2,400 genomes) completely finished in 3 days and 15 h. If this workflow was run sequentially on Crane, its cumulative running time would be 115 days and 18 h. On the other hand, ProkEvo with the *S.* Newport dataset finished in 7 days and 4 h when OSG was used as a computational platform. Similarly, if this workflow was run sequentially on OSG, its cumulative running time would be 1 year and 69 days. The longer runtime for the workflow on Crane vs. OSG is likely due to the variable resources of OSG and its different configurations and hardware. The nature of OSG also means that jobs may be preempted if a resource owner submits more jobs. In this case, the preempted job is retried, but that additional time is added to the workflow wall time. While the maximum number of independent jobs ran on Crane in one day is 2,377, this number is 8,606 when OSG was used. In general, HTC resources such as OSG are advantageous when a high number of jobs and nodes can be run and used simultaneously for workflows that can be completed efficiently, limiting

**Table 1 Comparison of existing pipelines for bacterial population genomics analyses.**

| Feature | ProkEvo | TORMES | Nullarbor | ASA³P | Bactopia |
|---|---|---|---|---|---|
| Sequence technology | Illumina | Illumina | Illumina | Illumina, PacBio, ONT, hybrid | Illumina |
| Paired-end reads | Yes | Yes | Yes | Yes | Yes |
| Single-end reads | Yes* | No | No | Yes | Yes |
| Workflow | Pegasus WMS***** | Bash + R | Perl + Bash | Groovy | Nextflow***** |
| Resume if stopped | Yes | No | Yes | No | Yes |
| Scalability to run on distributed and cloud resources | Yes | No | No | Yes | Yes |
| Input setup | Input file containing list of SRA accessions or file with absolute paths to local FASTQ files | Input file with metadata information about the input reads and their location | Input "samples" file with information about the isolates and their system location | All input files and meta information should be within a dedicated directory | Input file containing sample name and absolute paths to input files |
| Bencharked on different computational platforms | Yes (cluster, grid, cloud) | Yes (laptop, computer) | No (computer) | Yes (laptop, cluster, cloud) | No (cluster) |
| Capability of adding new tools | Yes* | No | No | No | Yes* |
| Guidance for performing population-based analyses | Yes (custom Jupyter Notebook and R code) | No | No | No | No |
| **Analyses** | | | | | |
| Quality control | Yes | Yes | Yes | Yes | Yes |
| Assembly | Yes | Yes | Yes | Yes | Yes |
| Pan/core-genome | Yes | Yes | Yes | Yes | Yes |
| Phylogeny | No | Yes | Yes | Yes** | Yes |
| BAPS Clustering | Yes | No | No | No | No |
| Comparative analyses | Bult-in and separate | Built-in | Built-in | Built-in | separate |
| Reporting | Text | R Markdown | HTML | HTML5 | Text |
| **Software and databases** | | | | | |
| Auto installed | Yes | No | No | No | Yes |
| Adjustable parameters for individual programs | Yes | Yes | Yes | No | Yes |
| Modifying parameters for individual programs before or while the pipeline runs | Before | Before | Before | Before | Before |
| Command-line adjustable options | No | Yes | Yes | No | Yes |
| Github repository | https://github.com/npavlovikj/ProkEvo | https://github.com/nmquijada/tormes | https://github.com/tseemann/nullarbor | https://github.com/oschwengers/asap | https://github.com/bactopia/bactopia |

(Continued)

| Feature | ProkEvo | TORMES | Nullarbor | ASA³P | Bactopia |
|---|---|---|---|---|---|
| Container available | Yes | No | Yes | Yes | Yes |
| Package manager | Conda YAML | Conda YAML | Bioconda and Brew | | Bioconda |
| Supports data download from NCBI | Yes (Run accessions) | No | No | No | Yes (Experiment/ Assembly accessions) |
| Supports locally stored data | Yes | Yes | Yes | Yes | Yes |
| Test datasets*** | | | | | |
| Size of test dataset | • 2,392 *S.* Newport genomes<br>• 2,870 *S.* Infantis genomes<br>• 23,045 *S.* Typhimurium genomes<br>• 21,919 *C. jejuni* genomes<br>• 11,990 *S. aureus* genomes | • 10 Salmonella spp. genomes | • 18 *Listeria* genomes<br>• 6 *Yersinia* genomes<br>• 23 *Salmonella enterica* subspecies enterica Serovar Bareilly genomes | • 4 *L. monocytogenes* genomes<br>• 8 *E. coli* genomes<br>• 32 *L. monocytogenes* genomes<br>• 128 *L. monocytogenes* genomes<br>• 1024 *L. Monocytogenes* genomes | • 1,664 Lactobacillus genomes |
| Number of cores used for testing | • ~100 CPUs—cluster****<br>• ~25,000 CPUs—grid<br>• 32 CPUs—cloud | • 4 CPUs—computer<br>• 32 CPUs—laptop | • 1 CPU—computer | • 32 CPUs—cloud<br>• 20 nodes with 40 CPUs—cluster<br>• 4 CPUs—laptop | • 96 CPUs—cluster |
| Output multiple directories under project | Yes | Yes | Yes | Yes | Yes |

Notes:
* Both ProkEvo and Bactopia are written using WMS which allows users to add more tools to the pipelines.
** The authors of ASA³P report that the phylogenetic tree did not finish for the dataset with 1,024 genomes due to lacking memory capacities.
*** The information about the test datasets and used resources for the various pipelines was found in their respective papers and GitHub pages.
**** Crane is Linux cluster, having 548 Intel Xeon nodes with 16 and 36 CPUs. Due to the fair-share policy and priority, we observed ~100 utilized CPUs at the time.
***** While Nextflow is more commonly used for bioinformatics applications, Pegasus WMS has the best overall performance for efficiently utilizing the computational resources (*Heller & Ghahramani, 2005*).

disruption from resource availability. The use of "all available resources" is often a limitation for University clusters. The total number of successful jobs ran with ProkEvo for the *S.* Newport dataset was 9,281 jobs on Crane and 16,624 jobs on OSG. Due to the opportunistic nature of the OSG resources, a running job can be cancelled and retried again, thus enabling the higher number of jobs reported by OSG. A similar pattern of performance metrics was observed when ProkEvo was run with the larger *S.* Typhimurium dataset (~23,000 genomes). When run on Crane, ProkEvo with *S.* Typhimurium completely finished in 15 days and 22 h. If this workflow was run sequentially on Crane, its

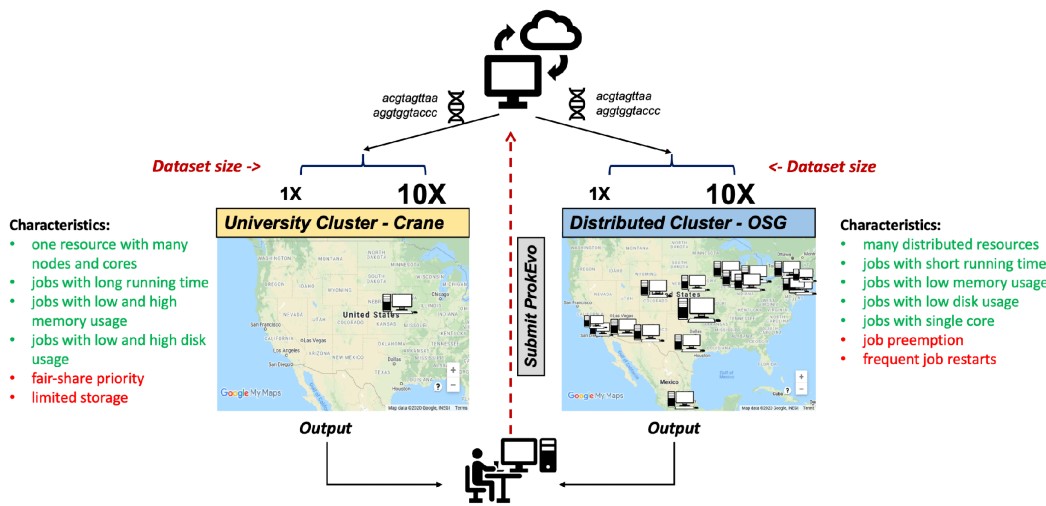

**Figure 3 Computational experimental approach to test the performance of ProkEvo using two different computational platforms with datasets of different size.** To test how ProkEvo would perform with a small (1X) vs. moderately large (10X) datasets, in addition to using different computational resources, we have designed the following experiment: (1) Selected two adequately sized datasets including genomes from *S.* Newport (1X–from USA) and *S.* Typhimurium (10X–worldwide); (2) Used two different types of computational platforms: Crane, the University of Nebraska high-performance computing cluster, and the Open Science Grid, as a distributed high-throughput computing cluster; (3) We then ran both datasets on the two platforms with ProkEvo, and collected the statistics for the performance in order to provide a comparison between the two different computational platforms, as well as possible guidance for future runs. Of note, the text in green and red correspond to advantages and disadvantages of using each computational platform, respectively. Map data ©2020 Google, INEGI.

**Table 2 Comparison of ProkEvo's performance on Crane and OSG with two datasets with significant difference in size and number of genomes.**

|  | Crane | OSG | Crane | OSG |
|---|---|---|---|---|
| **Number of genomes** | 2,392 |  | 23,045 |  |
| **Total distributed running time***  | 3 days 15 h | 7 days 4 h | 15 days 22 h | 26 days 6 h |
| **Total estimated sequential running time**** | 115 days 18 h | 1 year 69 days | 2 years 268 days | 13 years 5 days |
| **Maximum jobs ran in a day***** | 2,377 | 8,608 | 12,382 | 25,540 |
| **Total number of jobs ran** | 9,281 | 16,624 | 217,942 | 232,422 |
| **Output data size** | 131 GB |  | 1.2 TB |  |

Notes:
* Total distributed running time is calculated when many independent tasks are executed simultaneously while each of them is utilizing a single core. This is the default behavior of ProkEvo.
** Total estimated sequential running time is calculated when all steps from the pipeline are assumed to be run sequentially, on a single core.
*** The number of maximum jobs ran in a day depends on the type and length of the job, and is not linear, i.e., some tasks run faster than others which is directly dependent of the type of job being done.

cumulative running time would be 2 years and 268 h. On the other hand, the ProkEvo run for *S.* Typhimurium finished in 26 days and 6 h, when using OSG as a computational platform. Similarly, if this workflow was run sequentially on OSG, its cumulative running time would be 13 years and 50 days. The maximum number of independent jobs ran on Crane and OSG is 12,382 and 25,540 respectively. The total number of successful jobs ran with the *S.* Typhimurium dataset is 217,942 on Crane and 232,422 on OSG.

The running times of the individual bioinformatics programs contribute to the total running time of ProkEvo. Depending on the program and the job, the individual running time per genome can vary from few minutes (for tasks such as downloading data, quality trimming, filtering, BAPS clustering) to few hours (for tasks such as de novo assembly and annotation). The output files from "pegasus-statistics" that contain information about the average running time per job for the *S*. Newport and *S*. Typhimurium datasets are available on Figshare (https://doi.org/10.6084/m9.figshare. 13640639). Regardless of the running time of the individual bioinformatics tools used in the pipeline, one of the advantages of ProkEvo is its modularity and capability to scale and run all independent programs at the same time when the computational resources are available.

Although the workflow run time was better when Crane was used as a computational platform, it should be pointed out that OSG is more efficient for datasets where there are more jobs running simultaneously and, in our case, more genomes analyzed. As long as resources are available and no preemption occurs, workflows running on OSG can expect excellent performance. Notably, when run on OSG, ProkEvo utilized resources shared by thirty-four different facilities. Failures and retries are expected to occur on OSG, and their proportion may vary. From our experience, the number of failures and retries were encountered in ~0.3–30% of the total number of jobs. The OSG support staff is highly responsive to these issues, which can also be masked by a resilient and fault-tolerant workflow management systems like Pegasus WMS. All the data, intermediate and final files generated by ProkEvo are stored under the researcher's allocated space on the file system on Crane. Depending on the file system, it is possible that there are file count and disk space quotas. When large ProkEvo workflows are run, users should be aware that quotas on different clusters can be exceeded. On the other hand, due to the non-shared nature of the file system of OSG, intermediate files are stored on different sites, and exceeding the quotas is usually not an issue.

Both Crane and OSG are computational platforms that have different structure and target different type of scientific computation. All analyses performed with ProkEvo fit both platforms well. Thus, we provide an unambiguous comparison of both platforms and show their advantages and drawbacks when large-scale workflows such as ProkEvo are run.

## Applications

To demonstrate various applications of ProkEvo to showing a hierarchical-based population genomic analysis of different bacterial pathogens, we used publicly available datasets from three phylogenetically distinct species of pathogens, including the zoonotic pathogens *Salmonella enterica* and *Campylobacter jejuni*, and the human pathogen *Staphylococcus aureus*, which causes different diseases based on inter-human transmission or transmission through contaminated foods. While these datasets are likely to be inflated with clinical isolates and undersampling from other environments (e.g., animal or environmental), our analysis have the primary objective of emphasizing the utilities, approaches, and applications that can result from formal hierarchical-based population

data mining with ProkEvo, as opposed to formal research goals based on hypothesis-testing. To achieve our objective, we present independent case studies with these organisms that encapsulate some of the most generally useful approaches for studying bacterial populations. Some keystone concepts regarding bacterial population genetics and biology of each of these pathogens are described below. Notably, ProkEvo can be used with essentially any bacterial species with a few limitations: (1) the MLST program only works if the target bacterial species has an allelic profile present in the database, or is incorporated by the user; and (2) SISTR is designed specifically for analysis of Salmonella, but the program can be easily blocked out from the pipeline by the user.

## Overview of the population structure and ecology for *Salmonella enterica*, *Campylobacter jejuni* and *Staphylococcus aureus*

To understand utilities of ProkEvo and its intended purpose of providing hierarchical-based genotypic classifications and associated genomic content variation (i.e., loci), it is important that users/researchers are familiar with relevant aspects of the biology and the concept of population structure of target organisms (Fig. 4). In this report, we focus on three different species of foodborne pathogens, *Salmonella enterica*, *Campylobacter jejuni*, and *Staphylococcus aureus*, that are common worldwide (*Abebe, Gugsa & Ahmed, 2020*) but are evolutionarily quite distinct from one another and have very unique aspects of their population structures and biology.

The genus *Salmonella* is a member of the Phylum Proteobacteria and populations of these organisms can be found as common inhabitants of the gastrointestinal tract in a wide range of mammals, birds, reptiles, and insects, with these organisms often being transmitted to humans through contaminated animal products, vegetables, fruits, and processed foods (*Ferrari et al., 2019*). *Salmonella* comprises two primary species (*S. enterica* and *S. bongori*), which are believed to have diverged from their last common ancestor approximately 40 million years ago (*Fookes et al., 2011*). Worldwide, *S. enterica* is the most frequently isolated species from human clinical cases and from most environments. The *S. enterica* species comprise six genetically distinct sub-species, but the vast majority (>90%) of known human cases are caused by populations descending from a single sub-species, namely *S. enterica subsp.* enterica (lineage I). Even within *S. enterica* lineage I, tremendous genetic and phenotypic diversity exists, and such diversity is illuminated by the large number of sub-populations that are differentiated serologically (referred to as "serovars") by unique combinations of lipopolysaccharide molecules and major protein components of their flagella on their cell surfaces (the Kauffman–White scheme) (*Rowe & Hall, 1989*; *Achtman et al., 2012*). More than 2,500 serovars have been defined in *Salmonella enterica*. Serovars represent relevant biological units for epidemiological surveillance and tracking, because isolates belonging to the same serovar show much less variation with respect to important traits such as range of host species, survival in the environment, efficiency of transmission to humans, and virulence characteristics, than isolates from different serovars (*Alikhan et al., 2018*; *Achtman et al., 2012*).

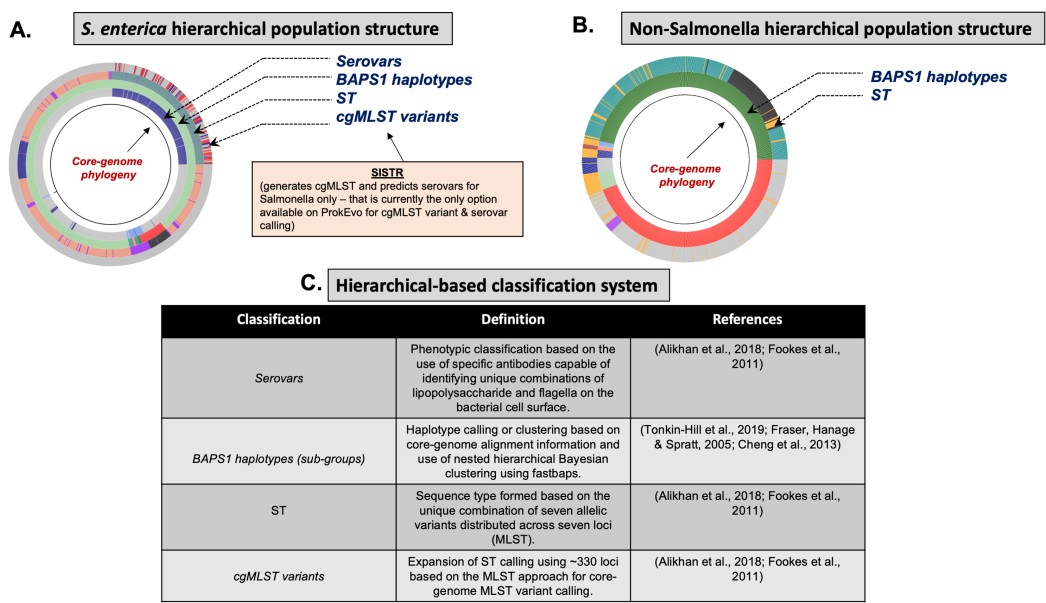

**Figure 4** **Demonstration of the hierarchical population structure mapping onto a core-genome phylogeny for Salmonella and non-Salmonella bacterial species, including specific clasifications.** (A) Salmonella hierarchical population structure follows a top-down ordering scheme: Serovars -> BAPS1 (level 1) haplotypes/sub-groups -> STs -> cgMLST variants (*Cheng et al., 2013*). (B) Non-hierarchical population structure includes only two layers of classification, including: BAPS1 (first), and ST (second). (C) Hierarchical-based classification system with related definitions is shown here with their respective references. Of note, prediction of serotypic information from genomic data, and cgMLST variant calling are only done for Salmonella in ProkEvo. Those are outputs specifically generated by SISTR which is specific for Salmonella. The user could either add another program that does it for other species, or use their metadata file containing such information.

Further evidence of the biological relevance of serovars comes from multi-locus genotyping and population genomics analysis across isolates from the Kaufmann–White-based serovars. These studies show that the serovars are covariates with the MLST-based population structure of *S. enterica* lineage I, with most serovars being found exclusively within a unique multi-locus Sequence Type (ST) or a clonally-related group of STs (ST clonal complexes). Consequently, the serotype of most isolates can be predicted accurately from MLST-based STs (*Achtman et al., 2012*).

The hierarchical, clonal structure of the *S. enterica* lineage I population can be visualized first by classifying genomes based on the serovar, and then increasing levels of resolution based on variation in seven-locus MLST and 330-locus cgMLST. Populations sharing alleles at the seven-loci MLST genotype are referred to as Sequence Types (STs), and members of an ST along with highly related STs (e.g., STs sharing alleles with at least 5 of 7 loci) are considered "clonal complexes". Genetically related variants at the cgMLST level are embedded within a single ST or group of clonally-related STs (*Alikhan et al., 2018*). In *S. enterica*, there are ~360 clonal complexes that are present across 50 of the most common serovars (*Alikhan et al., 2018*).

Within a serotype and individual ST, there are hundreds-thousands of different cgMLST variants. Although cgMLSTs derived from a single ST do share a common

ancestor, inferring the evolutionary relationships is computationally intensive and phylogenetic inference of thousands of cgMLST variants across more than one ST is not scalable computationally, particularly when it is necessary to account for horizontal gene transfer (HGT) and recombination events across divergent variants. To overcome this problem, phylogenetic relationships between multi-locus genotypic classifications can be inferred by combining with scalable, nested-clustering approaches using heuristic Bayesian-based computations such as BAPS, which determines genotypic relationships based on compositional features of the core-genome at different scales of resolution. Thus, evolutionary relationships of cgMLST variants within and between STs can be inferred efficiently through hierarchical classification of the genomes at six BAPS levels (BAPS1 being the lowest level, and BAPS6 the highest level of resolution and population fragmentation). BAPS1 through BAPS6 represent distinct levels, or stratum, of resolution used to portioning a population. Within each stratum, sub-groups or haplotypes are formed as discrete clusters that group core-genome sequences together, thereby generating a table with numerical identification for those clusters. Preliminary analyses showed that multiple STs can be part of the same sub-group within BAPS1, implying they have shared a common ancestor, and this context allows for evolutionary inference of cgMLSTs and their corresponding STs. Hierarchically combining Bayesian clustered-based genotyping schemes at low levels of resolution with ST and cgMLSTs has been shown previously in *Salmonella* (*Connor et al., 2016*), as well as other organisms such as *Enterococcus faecium* (*Moradigaravand et al., 2017*). Consequently, our heuristic-based approach uses the following hierarchical levels of population structure analysis for *Salmonella*: (1) Serovar; (2) BAPS1; (3) STs; and (4) cgMLSTs (Fig. 4).

In our study, we used the Serovar/BAPS/MLST/cgMLST classifications of genomes representing three serovars of *S. enterica* lineage I: *S.* Infantis, *S.* Newport, and *S.* Typhimurium. These are among the top twenty-five most prevalent zoonotic serovars of *Salmonella* according to the Center for Disease Control and Prevention (*CDC, 2019b*) but have distinct population structures and ecologies. While all three serovars are known for causing gastroenteritis in humans and have reservoirs in livestock, the bovine reservoir appears to be the most common source for *S.* Infantis and *S.* Newport, whereas *S.* Typhimurium has a generalist lifestyle and can be found in swine, poultry, bovine, etc. (*Ferrari et al., 2019*).

Taxonomically related to *Salmonella* at the Phylum level (Proteobacteria) is the genus *Campylobacter*, which includes two major species (*C. jejuni* and *C. coli*) that are frequent causes of gastrointestinal diseases in humans (*Sheppard & Maiden, 2015*). *Campylobacter* and *Salmonella* diverge taxonomically at the Class level (*Campylobacter* are members of the Epsilon class of Proteobacteria while *Salmonella* belongs to the Gamma Proteobacteria). Species of *Campylobacter* are also morphologically (helical cells) and physiologically (microaerophilic) distinct from *Salmonella*. However, like *Salmonella*, species of *Campylobacter* have reservoirs in food animals and often are associated with zoonotic outbreaks of foodborne illness in developed countries (*CDC, 2019a*).

*C. jejuni* can also be classified serologically by combinations of lipopolysaccharide and flagellar antigens, but far fewer serotypes are known for *C. jejuni* and serotyping is not

commonly included in routine diagnostic procedures. A distinguishing feature of *C. jejuni* population structure and evolutionary processes is its propensity for recombination and high frequency of HGT, which are mediated by specialized systems present in these organisms for uptake and recombination of extracellular DNA (*Sheppard & Maiden, 2015*). Consequently, *C. jejuni* is less clonal than any given serovar of *S. enterica* lineage I, and it contains a variety of widespread STs, for which the diversification patterns appear to be associated with host adaptation (*Sheppard & Maiden, 2015*; *Griekspoor et al., 2013*). Nonetheless, the hierarchical-based population approaches implemented in ProkEvo (which do not include serotype for this organism), are still able to associate STs with BAPS1 haplotypes to infer evolutionary relationships despite the genomic heterogeneity caused by higher levels of recombination (Fig. 4B).

While *Salmonella* and *C. jejuni* are divergent taxa within the Proteobacteria with gram-negative cell wall structures, the species *S. aureus* has a Gram-positive cell wall architecture and belongs to the phylum Firmicutes, which is evolutionarily very distant from the Proteobacteria. *Staphylococcus aureus* can cause a diverse array of diseases in humans including skin and invasive cutaneous infections, endocarditis, and toxic shock, but this organism is also recognized as a foodborne pathogen that causes foodborne intoxications (*Tong et al., 2015*). Foodborne gastroenteritis caused by this pathogen is due to the production of one or more heat-stable enterotoxins that are secreted during growth of the organism in contaminated foods (*Fetsch & Johler, 2018*). Humans are considered the most important reservoir of this organism, where it can be found on human skin, nasal cavities, and even the gastrointestinal tract, but the organism also colonizes similar anatomical sites in livestock. From WGS data, *Staphylococcus aureus* populations can be structured the same way as that of *Salmonella* and *C. jejuni* using BAPS1 and STs. However, this pathogen is not as diverse as *C. jejuni* at the ST level, but instead has a degree of clonality that is more comparable to those serovars within *S. enterica* lineage I. Because the organisms are not routinely serotyped, ProkEvo hierarchically classifies *S. aureus* genomes based on BAPS1 and MLST (Fig. 4B).

In this era of systems biology and multi-omics methodologies, it is becoming increasingly desirable to go beyond simple application of WGS for source-tracking and epidemiological investigation to understand dynamics of sub-populations of pathogenic species and the evolutionary and ecological characteristics that drive population disturbances. To study these dynamics and evolutionary/ecological processes, genotypic classifications of isolates (e.g., serovar, BAPS, MLST, cgMLST genotypic classifications) must be linked with important metadata (e.g., environmental/clinical source of the isolate, geography, etc.) as well as phenotypic data (predicted or laboratory-determined) such as resistance to antimicrobial agents, virulence characteristics, host adaptation, environmental survival. The linked genotypic and phenotypic sets the stage for quantitative genomics approaches to associate variation at specific genomic loci with phenotypes that are driving evolutionary processes (selection) and ecological adaptation in animal and food production environments or transmission/virulence attributes in humans (*Cury et al., 2018*). Genes and pathways marked by these processes illuminate selective pressures and better inform risk assessments as well as development of strategies to

mitigate spread. Therefore, we designed the studies described below to provide a practical example of how to link the distribution of important loci such as AMR to the population structures of the three *S. enterica* lineage I serovars, *C. jejuni* and *S. aureus* (*Bawn et al., 2020*; *Mourkas et al., 2019*; *Holden et al., 2013*).

### Case study 1: S. Newport population structure analysis

The *S. enterica* serovar Newport ranks among the top 25 serovars considered as emerging pathogens by the U.S. Centers for Disease Control and Prevention due to several recent outbreaks of foodborne gastroenteritis in humans (*Schneider et al., 2011*). Unlike most serovars of *Salmonella enterica* lineage I, where global populations of the serotype are dominated by a single ST clonal complex, the *S. Newport serovar has diversified into four major STs (Fig. 5A). The genetic diversity detected in *S. Newport isolates is somewhat surprising given its relatively recent emergence as an important human pathogen, and hence, low representation among isolates from the USA available in the NCBI SRA database (total of 2,392 isolates). Nonetheless, this serovar provides a robust example for analysis of a moderately complex population structure through ProkEvo. After the pre-processing steps, assemblies of genomes from 2,365 isolates passed the filtering step with a total data output of 131 GB produced by ProkEvo. After filtering for potentially misclassified genomes using the output of SISTR, 2,317 genomes remained that were annotated as *S. Newport and predicted as *S. Newport genotypically (Fig. S1 and Fig. S2). Thus, SISTR-based serovar predictions suggest that 2.03% of the genomes were misclassified as Newport. Using the genotypes assigned by the MLST, cgMLST, and BAPS-based genomic composition programs implemented in ProkEvo, we next defined the relative frequency of each genotype among 2,317 isolates (Figs. 5A–5H). This analysis identified the expected structure with four dominant STs in the following descending order: ST118, ST45, ST5, and ST132. The cgMLST distribution identified a total of 764 unique cgMLST variants, with the cgMLST genotype 1468400426 representing the most frequent variant (Fig. 5B) that accounted for ~14% of all isolates, whereas the distribution of the other cgMLST variants nearly ranging from 0.04% to 4.5%.

Circumventing the difficulties of scaling phylogenetic inference from thousands of core-genome alignments, we next examined genetic relationships of cgMLST variants using the scalable Bayesian-based clustering approach with BAPS to define sub-groups/ haplotypes based solely on the core-genome composition at different scales of resolution. As expected, BAPS-based haplotypes at increasing levels of resolution (BAPS1–BAPS6) increasingly fragmented the *S. Newport into: 9 sub-groups for BAPS1, 32 sub-groups for BAPS2, 83 sub-groups for BAPS3, 142 sub-groups for BAPS4, 233 sub-groups for BAPS5, and 333 sub-groups for BAPS6 (Figs. 5C–5H). We next used a hierarchical analysis to group the *S. Newport cgMLST variants and their STs based on shared genomic content at BAPS level 1 (BAPS1). At BAPS1 there are 9 total haplotypes or sub-groups. This analysis showed that the dominant BAPS1 haplotype (BAPS1 sub-group 8) is shared by two of the dominant STs, ST118 and ST5 (Fig. S1A). The shared BAPS haplotype implies that the clonal complexes defined by these dominant STs are more related to each other than to ST45 or ST132, which is consistent with the genetic relationships of these STs predicted by

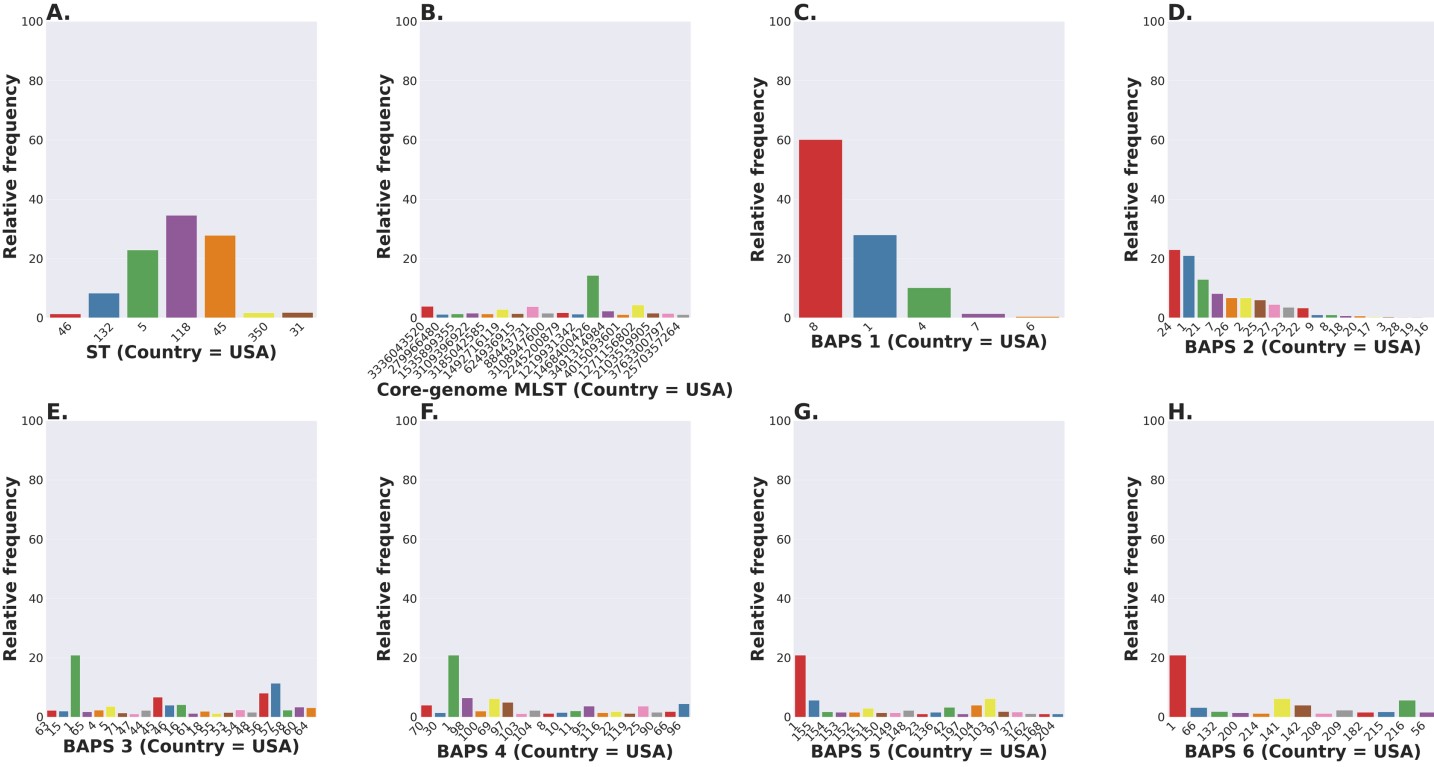

**Figure 5** *Salmonella* **Newport (USA) population stratification by genotype classification using two methods: allelic calls (ST and cgMLST) and a heuristic Bayesian approach (BAPS).** (A) ST distribution based on seven ubiquitous and genome-scattered loci using the MLST program, which is based on the PubMLST typing schemes (plot excludes STs with relative frequency below 1%). (B) Core-genome MLST variant distribution based on SISTR which uses ~330 ubiquitous loci (plot excludes STs with relative frequency below 1%). (C–H) BAPS levels 1-6 relative frequencies. For BAPS levels 3-6, we have excluded sub-groups that were below 1% in relative frequency in order to facilitate visualization. Within each BAPS level (1 through 6), each number represents a distinct cluster, or sub-group, to which the isolates belong to. The initial number of genomes used as an input was 2,392, while these analyses were run with 2,365 genomes that passed the post-assembly filtering steps.

e-BURST (*Alikhan et al., 2018*). Of note, there was not a single dominant cgMLST within any of BAPS1 sub-group 8 STs (ST118, ST5, or ST350) (Figs. S1B–S1D), but instead relatively large numbers of cgMLST variants were partitioned among these STs with 307, 149, and 23 cgMLST variants descending from the ST118, ST5, and ST350 clonal complexes, respectively. The association of a large number of cgMLST variants with ST118 suggests this population is rapidly diversifying, but it is important to note that the diversity within ST118 may be inflated compared to other STs based on sample bias and size.

In contrast to the genetic relationships of ST118, ST5, and ST350 found among BAPS1 sub-group 8, we also found that ST45 belongs to a distinct BAPS1 haplotype (sub-group 1) (Fig. S1E), with a total of 152 cgMLST variants. The ST45 complex also included a high frequency cgMLST variant, cgMLST 1468400426, which is the most dominant one for the entire *S.* Newport dataset (Fig. S1F). This predominance of cgMLST 1468400426 (within ST45 and across STs), could be due several reasons, including, but not limited to: (1) sampling bias; (2) recent outbreaks; (3) founder effect; or (4) a true selective advantage. Selection and founder effects often underlie the emergence of epidemiological clones

that cause significant increases in the numbers of outbreaks (*Grad et al., 2012*; *Fraser, Hanage & Spratt, 2005*).

After identifying the dominant cgMLST variant 1468400426, we next examined the degree of genotypic homogeneity (i.e., clonality) in this variant, and compared its genetic relationship to all other cgMLSTs within BAPS1 and ST45 clonal complex (Figs. S2A–S2E). We used BAPS-based groupings to estimate genetic relationships of the cgMLSTs to one another. This was accomplished by comparing the frequency of cgMLSTs within BAPS sub-groups at increasing levels of BAPS-based nested partitioning (increasing resolution from level from 2 to 6). To visualize partitioning of the cgMLST variants, genomes belonging to BAPS1 sub-group 1 and ST45 were first selected, and then each was categorized into two groups: one group containing cgMLST 1468400426 (numbered 1), and a second group contained all other cgMLSTs (numbered 0). This was done for each genome at each successive level of BAPS2-BAPS6. If the dominant cgMLST 1468400426 is highly clonal, it will be present in one or only a few of the BAPS sub-groups at each level of BAPS resolution. As shown in Figs. S2A–S2E, we found that indeed genomes belonging to the dominant cgMLST 1468400426 variant were all found within a single BAPS sub-group, even at the highest level of resolution (BAPS6). Notably, at each BAPS level, there are other cgMLST variants that also co-mapped to the same BAPS sub-groups as the dominant cgMLST 1468400426 (sub-group 1 at each level from BAPS2-6— matching colors between the two stacked bar plots), and the frequency of these other cgMLST variants that share BAPS with the dominant cgMLST 1468400426 is essentially stable as the BAPS resolution increases. These shared BAPS sub-groupings by cgMLST 1468400426 and other co-major variants including cgMLST 2245200879, cgMLST 843553928, cgMLST 3650140337, cgMLST 4212442350 (in addition to another forty-five minor cgMLST variants) at different levels, suggest a recent common ancestry, and illustrate how a nested-clustering approach such as BAPS can be used to infer evolutionary relationships in a scalable fashion.

While the above analysis defined population stratification within BAPS1 and ST45 clonal complex, further analysis of cgMLSTs among ST3045, ST3494, ST3783, and ST4493 showed that cgMLST 1468400426 is rarely found within ST3045 and ST4493; with only one genome of this cgMLST found in each of these two STs. Such a pattern is consistent with the BAPS-based relationships of ST3045, ST3494, ST3783, and ST4493, because these STs all belong to the same BAPS1 sub-group 1 along with ST45. Collectively, this hierarchical analysis of the genomic relatedness of ST and dominant cgMLST variants provide a systematic way to understand population structure and evolutionary relationships of cgMLST variants without the need for computationally intensive phylogeny. All the steps of these analyses are publicly available in our Jupyter Notebook (https://github.com/npavlovikj/ProkEvo/blob/master/jupyter_r_notebooks/salmonella_newport_analysis.ipynb).

### Case study 2: S. Typhimurium population-based analysis

*S.* Typhimurium is the most widespread serovar of *S. enterica* worldwide (*Sun et al., 2020*). Its dominance is partially attributed to its inherent capacity to move across a variety of

animal reservoirs including poultry, bovine, swine, and plants, and ultimately its zoonotic potential with propensity to cause gastroenteritis or non-Typhoidal Salmonellosis (*Crump et al., 2015*; *Ferrari et al., 2019*). This serovar is phenotypically divided into biphasic and monophasic sub-populations based on their expression of major flagellin proteins, for which both of the major flagellin genes are expressed in biphasic sub-populations; whereas, only one of the flagellin genes is expressed in monophasic sub-populations (*Sun et al., 2020*). Monophasic *S.* Typhimurium is an emerging zoonotic sub-population and isolates are often resistant to multiple drugs and heavy-metals (copper, arsenic, and silver) (*Sun et al., 2020*; *Branchu et al., 2019*; *Arai et al., 2019*). Due to its relevance as a major zoonotic pathogen and its frequent isolation from clinical and environmental samples, *S.* Typhimurium genomes from a large number of isolates are available (23,045 genomes of isolates from various continents). The large number of genomes available from *S.* Typhimurium dataset is a good measure of the scalability of ProkEvo, since it is an order of magnitude larger than *S.* Newport in the number of genomes. The geographical location of isolates from which these genomes were obtained cannot be determined uniformly from any single field of the associated metadata deposited to NCBI SRA, thus we focus only on scalability, and relationships of ST clonal complexes and cgMLST variants to one another. After the download and the pre-processing steps, 21,534 assemblies passed the filtering step, yielding a total data output from ProkEvo of 1.2 TB.

As with *S.* Newport, we also conducted an analysis of the population structure based on MLST and cgMLST. However, the sheer size of the *S.* Typhimurium dataset made it necessary to divide the 21,534 genomes into 20 different subsets, with genomes randomly assigned to each subset, in order to accomplish the following computational tasks: (1) pan-genome annotation and core-genome alignment with Roary; and (2) BAPS clustering using core-genome alignment across subsets. Nonetheless, it is important to reiterate that MLST, SISTR, and AMR genes identification using ABRicate are performed on the entire data set, since those tasks are carried out independently from Roary within ProkEvo. Because of the sub-setting of the *S.* Typhimurium dataset, the BAPS-based inquiry of each individual subset precludes direct comparison of BAPS distributions from each subset because of the varying numbering system of sub-groups. Therefore, BAPS1 level analysis is not presented in here.

After quality controlling and filtering the data, the dataset comprised 20,239 genomes of *S.* Typhimurium biphasic and monophasic isolates. In order to present various ways of conducting population-based analyses using ProkEvo, we use combinations of three pieces of information: (1) whether or not the genome is classified as biphasic or monophasic based on the SISTR algorithm (.csv); (2) the ST clonal complex calls using the legacy MLST (.csv); and (3) the cgMLST variant classification based on SISTR (.csv) (*Ferrari et al., 2019*). It is important to note that SISTR makes predictions of serotypes based solely on genotypic information. In *Salmonella* that is possible, because of the high degree of linkage disequilibrium between the clonal-frame (i.e., genome backbone) and loci that generate the O and H antigens (*Moradigaravand et al., 2017*). In the *S.* Typhimurium dataset, 72.6%, 25%, 2.4% of the quality-controlled genomes were classified as Biphasic,
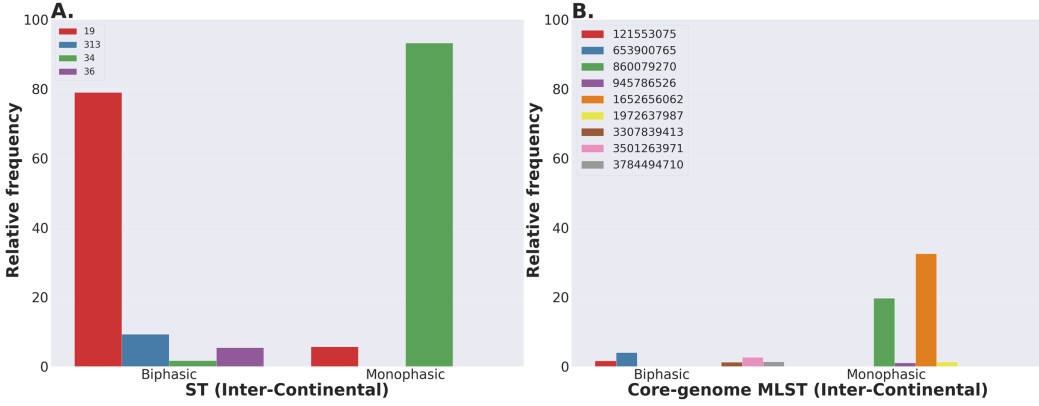

**Figure 6 Inter-continental distribution of *Salmonella* Typhimurium STs and core-genome MLSTs.**
(A–B) Relative frequencies of STs and core-genome MLSTs between Monophasic and Biphasic populations across multiple continents (STs and core-genome MLST variants with proportion below 1% were excluded from the graph). The initial number of genomes used as an input was 23,045, while these analyses were run with 21,534 genomes that passed the filtering steps. Raw sequences were downloaded from NCBI SRA without filtering for USA isolates exclusively. Hence, the name "Inter-Continental". However, we cannot break the data down into continents, because the metadata was unreliable. Bars that are not visible for a particular ST or cgMLST variant are either completely absent for that group, or present in a miniscule relative frequency.

Monophasic, or other serovars, respectively. From the Biphasic population, 78.4%, 9.62%, 5.35%, 2.09% of the isolates belonged to ST19, ST313, ST36, and ST34, respectively (Fig. 6A). Whereas, for Monophasic, 93%, 5.79%, 0.094% of the isolates belonged to ST34, ST19, and ST36, respectively (Fig. 6A). This partitioning matches the known clonality of the *S.* Typhimurium Monophasic populations and their association with the ST34 clonal complex (*Bawn et al., 2020*). As for the Biphasic population, it is predominantly associated with ST19 and likely contains the ancestor of the other ST clonal complexes. Most likely, the ST19 dominance is a consequence of its dispersal and unique adaptive traits which enabled its spread across a variety of animal and environmental reservoirs (*Bawn et al., 2020*). The much lower frequency ST313 and ST36 are associated with non-Typhoidal Salmonellosis and gastroenteritis in humans, and may be less adapted to environmental dispersal (*Bawn et al., 2020*).

In terms of cgMLST genotypic distributions, Biphasic and Monophasic had 5,162 vs. 1,161 unique cgMLST variants, respectively. This is expected since the dataset for Biphasic (~75% of the genomes) was larger than Monophasic (~25% of the genomes) and the population is presumably "older" based on the recent emergence of Monophasic isolates. The Biphasic population had much greater diversity, with many low-frequency cgMLST variants, and no individual cgMLST variant clearly dominating the population. This pattern is expected in a diversifying global population that has not recently experienced a selective sweep. In contrast, the Monophasic population comprised 1,161 total cgMLST genotypes, but two cgMLST variants (cgMLST 1652656062 and cgMLST 860079270) comprised 32.33% and 19.62% respectively, of the total number of isolates (Fig. 6B). This dramatic difference in frequency distribution of individual Monophasic cgMLST variants could be a consequence of, but not limited to, the following factors: (1) oversampling

of recent outbreaks; and/or (2) recent population bottlenecks or founder effect. All the steps for this analysis are shown in our Jupyter Notebook (https://github.com/npavlovikj/ ProkEvo/blob/master/jupyter_r_notebooks/salmonella_typhimurium_analysis.ipynb).

### Case study 3: Distribution of known AMR loci across the population structures of S. Infantis, S. Newport, and S. Typhimurium

In case study 3, we illustrate the use of ProkEvo to define the distributions of known AMR conferring loci from the Resfinder database of ABRicate across populations of *S. enterica* lineage I (*S.* Infantis, *S.* Newport, and *S.* Typhimurium described above). The goal of this analysis was to show how relationships between the population structures and the distribution of AMR loci can be identified. As described in the Methods section, population-specific results are provided by ProkEvo with ABRicate for several databases such as Resfinder, and the user may specify a database of interest, or may elect to use the ProkEvo option of reporting the results comparatively. Our emphasis here on results from the Resfinder database are driven largely by its broad applications to the fields of ecology and genomic epidemiology (*Perron et al., 2015*; *Cooper et al., 2020*). Although AMR phenotype predictions based on gene content alone do not have the same precision as measuring AMR phenotypes in the laboratory, monitoring AMR gene frequencies in specific populations of organisms does have the advantage of identifying potentially significant population-scale events that are relevant to public health (e.g., changes in frequencies or new combinations appearing within a population).

Results from the analysis with the Resfinder database identified 72 unique AMR loci in genomes of *S.* Infantis, 125 unique AMR loci in *S.* Newport, and 408 unique AMR loci in *S.* Typhimurium (Table S2). All 72 AMR genes in *S.* Infantis were confined solely to ST32, which could be associated with the acquisition of mega-plasmids carrying distinct combinations of AMR genes (*Aviv et al., 2014*; *Franco et al., 2015*). In contrast, large numbers of AMR loci were found in three of the major clonal complexes of *S.* Newport, with 57 AMR loci in ST118, 84 AMR loci in ST45 and 33 AMR loci in ST5. Similarly, large numbers of AMR loci were found among each of the four most dominant ST clonal complexes of *S.* Typhimirium; ST19 had the most with 301 AMR loci, ST34 had 249 AMR loci, ST36 had 130 AMR loci, and ST313 had 112 AMR loci. Given that ST19 and ST34 are the most frequent STs in the database for this serovar, it is not surprising that their repertoire of genes would be higher than the others (*Alikhan et al., 2018*; *Bawn et al., 2020*). Among the AMR genes identified from any of the three serovars, apparent orthologues of genes known to confer resistance to a broad range of antibiotic classes were identified, including tetracyclines (*tet* genes), sulfonamides (*sul* genes), macrolides (*mdf* genes), florfenicol and chloramphenicol (*florR* and *catA* genes), trimethoprim (*dfrA* genes), beta-lactamases (*bla* family of genes), and aminoglycosides including streptomycin and spectinomycin (*aph*, *ant*, *aadA*, and *aac* genes) (*McArthur et al., 2013*).

Although significant numbers of AMR genes were found in many of the dominant STs for each serovar, it should be noted that most of the AMR loci were sparsely distributed among small numbers of isolates within an ST (Table S2). Therefore, we used a threshold of presence of a given AMR gene in >=25% of the genomes of an individual ST clonal

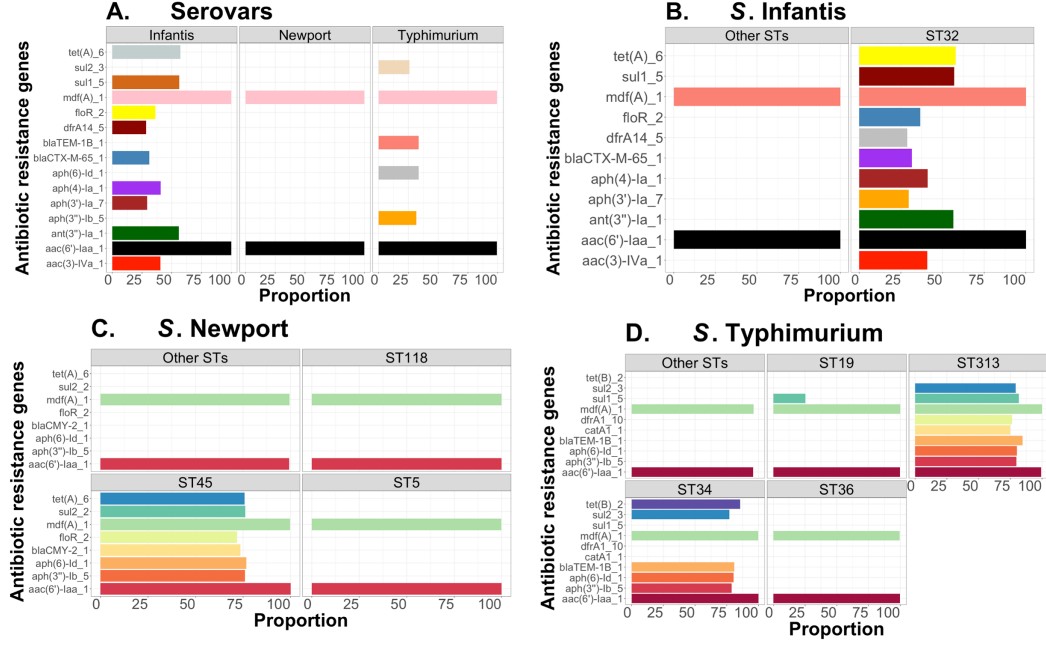

**Figure 7 Antibiotic-associated resistance genes distribution between and within three serovars of**
***S. enterica* lineage I.** (A) Proportion of genomes containing antibiotic-associated resistance genes
within each serovar. (B–D) Proportion of antibiotic-associated resistance genes within major vs. other
STs for *S.* Infantis, *S.* Newport, and *S.* Typhimurium, respectively. For the plots, (B–D), the population
was initially aggregated based on the dominant STs vs. the others, prior to calculating the relative fre-
quency of genomes containing each antibiotic-resistance gene. Only proportions equal to or greater than
25% (post-hoc threshold) are shown. For *S.* Infantis and *S.* Newport, only USA data were used; whereas,
for *S.* Typhimurium we did not filter based on geography in order to have a larger dataset to test Pro-
kEvo's computational performance. Datasets were not filtered for any other epidemiological factor.
The total number of genomes used for this analysis was 1,684, 2,365, 21,509 for *S.* Infantis, *S.* Newport,
and *S.* Typhimurium, respectively, after filtering out all missing or erroneous values. Also, there were 18
and 1,666 genomes for "Other STs" and ST32 within the *S.* Infantis data, respectively. For *S.* Newport,
there were 393, 800, 643, and 529 genomes of the following groups: Other STs, ST118, ST45, and ST5,
respectively. Lastly, for *S.* Typhimurium, there were 1,430, 12,477, 1,493, 5,274, and 835 genomes for
either Other STs, ST19, ST313, ST34, or ST36, respectively.

complex in order to define the predominant AMR patterns in each population.

As illustrated in Fig. 7, three major patterns were apparent. First, we note that the patterns
of predominant AMR genes were largely somewhat unique to each serovar (Fig. 7A).
Second, the largest numbers of predominant AMR genes were confined to individual STs
in *S.* Infantis (ST32) and *S.* Newport (ST45) and two of the four dominant STs (ST34 and
ST313) in S. Typhimurium (Figs. 7B–7D). This may reflect a higher degree of clonality
among these STs, but could also be an artefact of oversampling clinical isolates during
outbreaks without accounting for the overall environmental diversity. Finally, we note the
widespread distribution of the mdf(A)_1 and aac(6′)-Iaa_1 loci across all serovars and all
clonal complexes (Fig. 7A). Such a high degree of conservation suggests these elements
may have been acquired ancestrally, prior to diversification of these three serovars, as
opposed to recent independent acquisitions (*Cohan, 2019*). Also, it is important to
mention that we are not differentiating between genes present in chromosome vs.

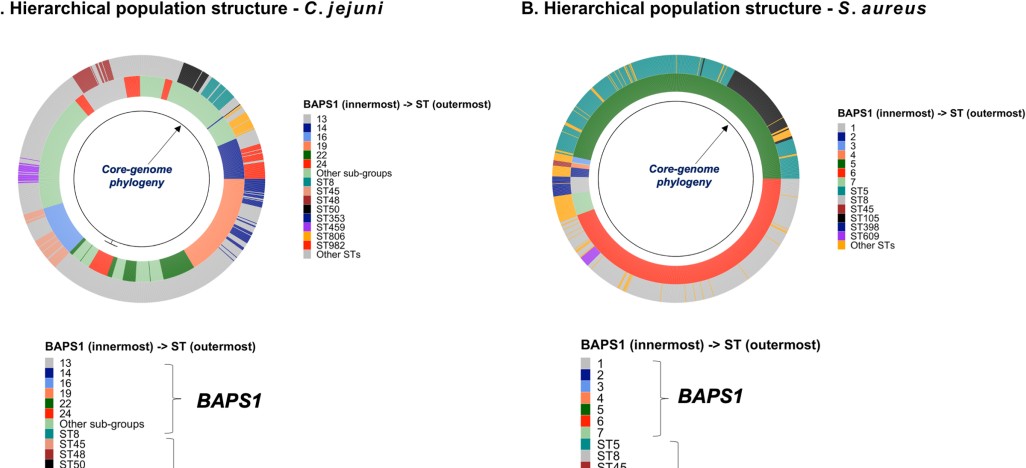

**Figure 8  Relationship between the core-genome phylogeny and population structure of *C. jejuni* and *S. aureus*.** (A–B) Population structure using BAPS1 and ST for genotypic classifications were overlaid onto the core-genome phylogeny (circle in black) of both *C. jejuni* and *S. aureus*, respectively. BAPS1 was used as the first layer of classification to demonstrate how each sub-group/cluster can be comprised of multiple STs. For instance, STs that cluster together, and belong to the same BAPS1 sub-group, are more likely to have shared a most recent common ancestor. This represents a hierarchical population-based analysis going from BAPS1 to STs. For this analysis and visualization, we have used a random sample composed of 1,044 and 1,193 genomes for *C. jejuni* and *S. aureus*, respectively.

plasmids. Plasmids are more promiscuous and facilitate HGT between closely related, or divergent populations (*Achtman & Zhou, 2014*).

## Case study 4: Population structures and distribution of AMR genes in *C. jejuni* and *S. aureus*

To further illustrate the versatility of ProkEvo across diverse microbial species, we examined the relationships of population structure and AMR gene distributions in two additional bacterial pathogens, *Campylobacter jejuni* and *Staphylococcus aureus*, which belong to very distantly related Phyla. In addition to its distinct morphology (helical) and physiology (microaerophilic) *C. jejuni* has a unique population structure (Fig. 8A) which features 23 major ST complexes whose evolutionary relationships are confounded the high frequency of gene acquisition and recombination (*Sheppard & Maiden, 2015*; *Griekspoor et al., 2013*; *Berthenet et al., 2019*; *Sheppard et al., 2013*).

*Staphyloccus aureus* is a Gram-Positive organism that belongs to the Phylum Firmicutes and is evolutionarily very distantly related to the Proteobacteria. Its population structure is highly clonal, and three STs (STs 8, 5, and 105) comprise more than 80% of the population of the species represented in the database (Fig. 8B). ST8 is associated with community-acquired infections in the form of either methicillin susceptible or resistant strains (MSSA or MRSA) (*Glaser et al., 2016*). ST5 can also cause skin infections and is often found as MRSA (*Baines et al., 2016*). ST105 appears to be closely related to ST5, and

isolates from both ST5 and ST105 can carry the *SCCmec* element II encoding broad-spectrum β-lactam resistance (*Challagundla et al., 2018*).

To define hierarchical relationships of major STs of these species, we used the BAPS1/ST classifications produced by ProkEvo. In addition to the bar charts previously demonstrated for the Salmonella datasets, we developed an integrative approach to visualize the frequencies and inter-relationships of STs and BAPS-based variants (Figs. 8A–8B), where each genome is depicted as a member of a circular phylogeny (central ring), and their coloration in concentric rings depicts their associated BAPS variants (innermost ring) and ST variants (outermost ring) for each genome. In this approach, position of each isolate on the ring is determined by phylogeny and groupings of the BAPS1/STs can be viewed in relationship to the inferred phylogeny. As shown in Fig. 8A, this visualization illustrates how genomes from individual ST complexes of *C. jejuni* are found within a single BAPS1 genotype. For example, the ST45 clonal complex is found exclusively within the BAPS1 sub-group 16, ST48 is confined within the BAPS1 sub-group 13, ST353 is found within the BAPS1 sub-group 19, and ST982 is found within the BAPS1 sub-group 14. Similarly, visualization of the relationships between BAPS1 sub-groups and the most frequent STs in *S. aureus* (Fig. 8B) illustrated the high degree of clonality in its population structure. The dominant ST5 and ST105 were found exclusively within the BAPS1 sub-group 5, ST398 was restricted to the BAPS1 sub-group 1, and the ST609 complex was found within the BAPS1 sub-group 6. Thus, despite the differences in population structure between species, the combination of Bayesian clustering (BAPS) and multi-locus genotyping in ProkEvo still enables detection and visualization of broad evolutionary relationships of the STs to one another, which might not be possible when attempting to scaling phylogenetic-based analysis.

Using the framework of hierarchical BAPS1-ST relationships, we next used ProkEvo outputs from ABRicate with the Resfinder database to examine distributions of AMR genes among the STs in these diverse organisms. For this analysis, we focused on STs representing >1% of the total number of genomes for both *C. jejuni* and *S. aureus* (Figs. 9A–9B). The ProkEvo-mediated search of the Resfinder database from *C. jejuni* genomes identified 256 unique AMR elements in *C. jejuni* and 164 AMR loci for *S. aureus*. Within *C. jejuni*, the top 8 most frequent STs had the following total number of AMR loci: ST353 (29), ST45 (30), ST982 (20), ST48 (24), ST50 (31), ST8 (20), ST806 (19), and ST459 (15). Thus, there was relatively even distribution of AMR loci among the most dominant STs. In contrast, the number of AMR loci in *S. aureus* was essentially a function of the frequency of the STs. The most frequent *S. aureus* STs in the database (ST8, ST5, and ST105) contained the largest number of AMR loci (ST8 had 88 AMR loci, ST5 had 85 AMR loci, and ST105 had 52 AMR loci). In contrast, the lower frequency STs had fewer (ST398 had 39 AMR loci, ST609 had 20 AMR loci, and ST45 had 24 AMR loci). The link to the intermediate files used to obtain this information can be found in Table S2.

As was the case with distributions of AMR loci in *Salmonella enterica*, most of the AMR loci detected in *C. jejuni* and *S. aureus* were sparsely distributed across isolates belonging to an individual ST. Therefore, we focused on AMR loci in >=25% of the isolates within an ST. As shown in Figs. 9C–9D, despite the fact that *C. jejuni* had a greater number

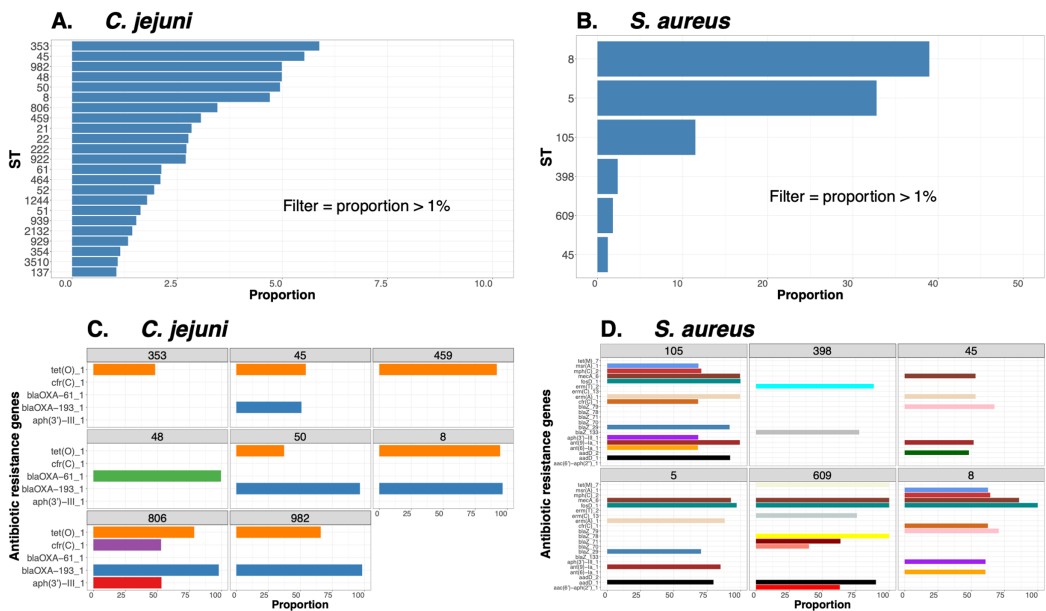

**Figure 9 ST-based population structure and distribution of antibiotic-associated resistance genes for two major foodborne pathogens.** (A–B) Proportion of the most dominant STs within *C. jejuni* and *S. aureus* populations (only proportions >1% are shown). (C–D) Proportion of genomes containing antibiotic-resistance genes within ST populations for *C. jejuni* and *S. aureus* (only proportions >25% are shown). Both datasets only included genomes from USA and were not filtered for any other epidemiological factor. The total number of genomes entered in this analysis was 18,845 and 11,597, for *C. jejuni* and *S. aureus*, respectively, after filtering out all missing or erroneous values. For *C. jejuni*, there were 886, 1,041, 940, 932, 1,108, 577, 651, and 940 genomes of the following groups: ST8, ST45, ST48, ST50, ST353, ST459, ST806, and ST982, respectively. Lastly, for *S. aureus*, there were 4,518, 3,801, 1,334, 276, 211, and 141 genomes for either ST8, ST5, ST105, ST398, ST609, or ST45, respectively.

of total AMR-associated genes, *S. aureus* had a greater number of prominent AMR loci meeting the >25% threshold, which might be associated with a higher proportion of clinical isolates in each dataset. The diversity of prominent AMR loci in *S. aureus* was also quite noticeable, with each of the major STs having a distinct combination of AMR loci.

With respect to *C. jejuni*, there was widespread co-occurrence of tet(O)_1 and blaOXA-193_1, which confer resistance to tetracyclines and beta-lactamases, respectively, across the five most frequent STs (Fig. 9C). One potential explanation for this pattern would be an ancestral acquisition of both genetic elements, and subsequent loss of one or both genes during divergence of the ST48, ST353, and ST459 clonal complexes (*Bobay & Ochman, 2018*). In contrast, the *cfr(C)_1* and *aph*(3′)-III_ loci appear uniquely in the *C. jejuni* ST806 clonal complex, suggesting these genes are relatively recent acquisitions within the clonal complex. The *cfr* gene is of great interest because it has a pleiotropic phenotype associated with resistance to a variety of AMR classes, such as: phenicol, lincosamide, oxazolidinone, pleuromutilin, streptogramin A, and other macrolides (*Atkinson et al., 2013*). Given those findings, we broadened the scope of the analysis by examining how the entire accessory genome distribution, and not just a fraction of it such as AMR genes, associated with the population structure using BAPS1 and ST clonal

complexes mapping onto it. To accomplish that, we used a random subset of 1,044 *C. jejuni* genomes—the same sample used to produce Fig. 8A. Roary, within ProkEvo, generates a dendrogram that is derived from a binary matrix of gene presence and absence from accessory genes only (accessory_binary_genes.fa.newick). We use this .newick file in addition to a .csv file containing the BAPS1 and ST clonal complexes information to produce the final plot with phandango. Our expectation was that if ST complexes are highly clonal (i.e., high degree of linkage disequilibrium between loci), their accessory-genome based distribution would be quite comparable, if not identical, to that of when plotted using the core-genome phylogeny, which contains the clonal-frame. Also, if BAPS1 sub-groups have recently shared a common ancestor, one would expect that more related sub-groups would cluster near each other in this analysis. In general, the data depicted in Fig. S3 indicate that related clonal complexes are more likely to share accessory loci, which could be explained by vertical transmission and/or HGT. One exception was ST459, which instead of only forming a discrete cohesive cluster based on the accessory loci distribution, appears to contain two sub-populations within it.

In the case of *S. aureus*, the largest number of prominent AMR genes were found in ST5 and ST105 (Fig. 9D), both of which belong to the same BAPS1 sub-group 5 genomic type, and are thus more closely related to each other than the other dominant STs (Fig. 8B). ST8 and ST609 also carry significant numbers of prominent AMR loci and these STs also share evolutionary history, since they both belong to BAPS1 sub-group 6 genomic type (Fig. 8B). In contrast, ST398 and ST45 contain the fewest AMR loci and each belongs to a distinct BAPS1 sub-group (ST398 is a member of BAPS1 sub-group 1 while ST45 is a member of BAPS1 sub-group 4—Fig. 8B). Similarly, to the accessory-genome based analysis performed for *C. jejuni*, we applied the same approach for a sample of 1,193 genomes of *S. aureus* (those present in Fig. 8B). Given that *S. aureus* is more clonal than *C. jejuni*, our data supports the hypothesis of a high degree of linkage between the clonal-frame and accessory genome, reflected by the discrete clusters formed by the conjunction of BAPS1 and ST clonal complexes (Fig. S4).

One limitation we have identified while plotting core-genome phylogenies combined with population structure data of both *C. jejuni* and *S. aureus*, was to achieve a high degree of resolution to visualize branching patterns. The black centered circle depicted in Figs. 8A–8B represents a core-genome phylogeny generated with FastTree. As it can be seen, branching patterns are not recognizable while using ggtree to plot with our configurations. Although we have attempted to change parameters, the branch length remained unchanged. Therefore, we tested whether we could improve visualizations by doing two things: (1) downsampling the data using smaller sample sizes of randomly selected genomes evenly distributed across major STs (i.e., accounting for population structure); and (2) by comparing how ggtree vs. phandango would enhance our plotting capabilities. In brief, we have found that downsampling using our approach did not improve ggtree phylogenetic plotting resolution. However, phandango enhanced phylogenetic visualization regardless of the sample size (Figs. S5–S14). Of note, ggtree has important advantages such as automation and high degree of control over figure aesthetics.

We suggest users to examine these as well as other related programs to accomplish this task.

## DISCUSSION

The continuous increase in the volume of WGS data from bacterial species is driving the field of bacterial genomics away from simple comparative and functional genomics towards population-scale inquiry. This shift requires approaches rooted in data science to process, analyze, and mine WGS data at scales that have not before been achieved. Indeed, the vast number of genomes currently available is already driving development of tools, pipelines, and approaches for analysis of population dynamics, phylogeography, and epidemiological patterns at whole genome-scales of resolution. When scalable tools for population-based inquiry at genomic resolution are combined with appropriate sampling of environments and robust metadata, these unique approaches will collectively provide entirely new ways to understand fundamental ecology of important microorganisms, environmental factors that drive ecological adaptation, and the evolutionary mechanisms through which such adaptations are mediated (*Sheppard, Guttman & Fitzgerald, 2018*; *Joseph & Read, 2010*; *Alikhan et al., 2018*; *Yahara et al., 2017*; *Power, Parkhill & de Oliveira, 2016*).

Scalability of phylogeny and hierarchical-based population classifications remain as key bottlenecks that limits population-based inquiry at genomic resolution. Automation and parallelization of complex pipelines for implementation on different types of computational platforms (e.g., clusters and grids) can help overcome the scalability bottleneck. ProkEvo fills this gap by allowing researchers to scale and automate the analyses from hundreds to many thousands of genomes without the need to write individual scripts to run programs and move data input/output from program to program. Indeed, such approaches become difficult to reproduce across laboratories. ProkEvo takes advantage of a set of well-developed bioinformatics tools and a robust workflow management system that enables execution on different high-performance and high-throughput computational platforms. Thus, ProkEvo produces scalable and highly reproducible workflows. We acknowledge that full automation of workflows in ProkEvo has trade-offs, because users may rely on such systems without understanding how the underlying assumptions and tuning of important parameters in the individual programs ultimately affect their studies. On the other hand, a large number of microbiology laboratories can immediately benefit from the automation and scalability of ProkEvo to generate a variety of novel hypotheses. Moreover, implementation of ProkEvo in these research environments will ultimately become a major force to drive development of more systematic approaches to study designs for large-scale genomic studies and ongoing surveillance, including the sampling collection of WGS from isolates, as well as the collection and curation of critical metadata.

ProkEvo is modular, and each genome is analyzed independently when computing resources are available. In theory, if a dataset has $n$ genomes and a computational platform has $n$ available cores, ProkEvo can easily scale linearly and utilize all resources at the same time on execution platforms such as clusters and grids. ProkEvo only needs a list of

NCBI SRA (genome) identifications as an input, and the Pegasus submit script. ProkEvo also works with genomes that are already locally available. The computational resources used for the steps in ProkEvo are specified per tool and are not fixed. This is an important feature of ProkEvo that allows efficient allocation of resources and requires high resources only when needed. While the scripts for executing the tools in ProkEvo are written to consider common errors, such as low-quality input data or exceptions, failures due to rare cases are still possible. In such instances, only a failed job is retried, with the possibility of terminating only the failed job upon repeated failure. Failure of individual jobs does not affect the continuity of the pipeline; instead, the remaining independent jobs continue running. This feature is extremely useful when analyzing large datasets, and bypasses the problem of very small fractions of the tens of thousands of genomes having faulty reads that would otherwise disrupt the entire workflow across all jobs.

Most of the enabling capability of ProkEvo relies upon automation and management of the massive workflows through Pegasus WMS. The scalability, ability to handle large sets of data with complex input/output dependencies, resource management, flexibility to add and remove programs, and portability to different computational platforms are just a few of the advantages that drove selection of Pegasus as the WMS for ProkEvo. Although we used ProkEvo in this report to efficiently process and analyze a moderately large dataset of >20,000 genomes from *Salmonella* Typhimurium, future testing needs to be done to evaluate and improve ProkEvo's performance with hundreds of thousands of genomes. Additionally, its portability to cloud environments such as the Amazon Web Service needs to be evaluated.

Despite the efficiency of the Pegasus WMS, one of the central programs of ProkEvo (Roary) creates a bottleneck in generating core-genome alignments. While this limitation might be particular to our settings and datasets, this step is important since it precedes population structure analysis using fastbaps or downstream phylogeny, and it can run indefinitely when the number of genomes is large. Our workaround here was to randomly divide the dataset into subsets of up to 2,000 genomes, which allows ProkEvo to perform all jobs efficiently. However, this approach has consequences because: (1) fastbaps uses Bayesian BAPS computations which may confound direct data aggregation afterwards; (2) the user will have to generate multiple phylogenetic trees; and (3) pan-genome annotation may vary across subsets and there may be inconsistent gene calls/ classifications, particularly with respect to hypothetical proteins. We are examining other scalable computational approaches for phylogenetic inference such as downsampling based on population structure and metadata, in addition to utilizing kmer-based construction of distance matrices using raw reads directly (*Ondov et al., 2016*).

An advantage of ProkEvo is that the Pegasus WMS can easily accommodate addition of novel programs/algorithms to the platform without disrupting any pre-established tasks. Hence, new solutions or alternative steps or programs can easily be incorporated into ProkEvo. Nonetheless, ProkEvo is expected to scale quite efficiently to many thousands of genomes to complete the following tasks, beginning with raw sequences: (1) MLST and SISTR classifications; (2) AMR, virulence, and plasmid mapping; and (3) Genome annotation with Prokka.

Pegasus WMS has been used for development of small and large-scale processing and computational pipelines for a variety of projects and applications across multiple disciplines, including LIGO gravitational wave detection analysis (*Usman et al., 2016*), the structural protein-ligand interactome (SPLINTER) project (*Quick et al., 2015*), the Soybean Knowledge Base (SOyKB) pipeline (*Liu et al., 2016*), and the Montage project for science-grade mosaics of the sky (*Berriman et al., 2004*). While Pegasus is a versatile WMS, other WMS such as Nextflow (*Di Tommaso et al., 2017*) and Snakemake (*Koster & Rahmann, 2012*) are more commonly used for bioinformatics applications. However, compared to the other WMS, Pegasus has the best overall performance for efficiently utilizing the computational resources (*Larsonneur et al., 2018*). Moreover, Pegasus WMS provides a unique robust support of multiple computational platforms, varying from publicly available clusters to distributed cloud and grid infrastructures (*Mitchell et al., 2019*).

To the best of our knowledge, use of an advanced WMS such as the Pegasus WMS is a very unique feature of ProkEvo that is not found in other complex pipelines for large-scale analysis bacterial genomes such as EnteroBase (*Zhou et al., 2019*), TORMES (*Quijada et al., 2019*), Nullarbor (*Seemann et al., 2020*), and ASA$^3$P (*Schwengers et al., 2020*). EnteroBase is an online resource for identifying and visualizing bacterial species-specific genotypes by utilizing a high-performance cluster at the University of Warwick. TORMES is a whole bacterial genome sequence analysis pipeline that works with raw Illumina paired-end reads, and is written in Bash. Nullarbor is a Perl pipeline for performing analyses and generating web reports from sequenced genomes of bacterial isolates for public health microbiology laboratories. ASA$^3$P is an automated and scalable assembly annotation and analyses pipeline for bacterial genomes written in Groovy. Bactopia is one of the most comprehensive pipelines available for analysis of bacterial genomes using Nextflow workflow manager (*Petit & Read, 2020*). Some of the main similarities and differences between the aforementioned pipelines are shown on Table 1. Because EnteroBase is a service where researchers upload data and get the desired outputs without any control over the tools and the parameters used, we omitted EnteroBase in the comparison in Table 1. While all of the remaining pipelines provide very similar types of analyses, the diversity of the analyses and the tools incorporated depends on the end-goal of the research group developing the pipeline. Both ProkEvo and Bactopia are written using WMS which allows users to add more tools to the pipelines. The documentations for Pegasus and Nextflow provide examples of how to do this. Adding new tools to TORMES, Nullarbor and ASA$^3$P is feasible, but not as easily manageable due to the way these pipelines are written (e.g., having one file with code for the whole pipeline that requires advanced programming knowledge). To the best of our knowledge, all the modifications of the programs and parameters used in all of these pipelines need to be done before running the pipeline itself. Nextflow, the WMS used for Bactopia, generates many intermediate files that can exceed the available storage quotas on the standard computational platforms. Pegasus, the WMS used for ProkEvo, supports clean-up of the intermediate files as the respective task finishes, which minimizes the possibility of exceeding the storage resources. Both ProkEvo and Bactopia install all the dependencies and databases as part of the pipeline. On the other hand, TORMES, Nullarbor and ASA$^3$P

require additional steps to set the necessary environments with the required tools and databases. ProkEvo and Bactopia support data download from NCBI, while the remaining pipelines do not. With ProkEvo we provide a set of custom Jupyter Notebook and R codes used for extracting meaningful information from the produced ProkEvo output. ProkEvo and ASA$^3$P are tested on multiple platforms, such as high-performance and high-throughput clusters and cloud. Nullarbor and Bactopia do not provide benchmarking on different computational platforms, while TORMES has been tested on a laptop and computer. Based on the information provided in the respective papers and GitHub repositories, to the best of our knowledge, ProkEvo is the only available pipeline that was successfully tested on datasets from ~2,400 to ~23,000 genomes each. The datasets used for testing TORMES and Nullarbor are in the range of 6–23 genomes, while ASA$^3$P and Bactopia were tested with 1,024 and 1,664 genomes respectively. Being able to perform genomics analyses on populations scalable to the 20-fold larger datasets than the ones presented with the other pipelines is a tremendous advantage of ProkEvo extremely important for researchers working in the field of population genomics. In addition to this, ProkEvo is also distinct in its ability to: (1) process each genome independently and utilize as many computational resources as possible; (2) efficiently utilize distributed, high-throughput computational platforms, such as OSG, with tens of thousands of available cores; (3) set memory and run time resources per bioinformatics tool and job and increment these values on retry; (4) combine classifications of each genome based on multi-locus genotypes (at ST and cgMLST scales) with the scalable approach of classifying genomic types based on Bayesian haplotype clustering using a nested-approach.

As illustrated in all four case studies, our approach of combining hierarchical combinations of genotypic classifications with relevant loci such as AMR genes, or even the entire accessory genome, can produce novel insights. In these case studies, we demonstrated how the combination of multi-locus approaches and Bayesian haplotype clustering analysis can illuminate evolutionary relationships with scalable methods. Our studies also identified combinations of AMR genes that are widely dispersed across dominant STs as well as AMR genes with population-specific patterns of distribution. Integrating population genomics (allele and clone frequencies) outputs from ProkEvo with complex trait analyses can begin to identify putative casual variants that are driving evolution and ecological characteristics (*Azarian, Huang & Hanage, 2020*). Population-based selective sweeps (i.e., purged genomic variation at the whole genome level) can be driven by acquisition of a single locus capable of providing novel physiological or virulence trait (*Cohan, 2019*), as exemplified by acquisition of novel loci in clonal complexes ST21 and ST45 of *C. jejuni* which reduce oxygen sensitivity and enhance survival and spread across the poultry food chain (*Yahara et al., 2017*). However, complex traits such as ecological fitness can also arise from contributions of allelic variation at multiple loci (*Sheppard, Guttman & Fitzgerald, 2018*; *Yue & Schifferli, 2014*). Of course, this is not a one-way street as such variation in bacterial pathogens is also met with variation in host loci that contributes to susceptibility, for instance (*Wang et al., 2018*).

It is important to note that our case studies drew upon the large numbers of genomes already available in the NCBI SRA databases, and we have been careful to draw only general features as an example of such analysis, because of the inherent bias in broad species or serovar-specific datasets. The most common bias in WGS representations of pathogenic species results from overrepresentation of clinical samples in general in addition to the potential oversampling large numbers of isolates from outbreaks and epidemiological variants. Such representation does provide important temporal approaches for detecting frequency changes in dominant variants, which even at high levels of resolution are indicative of significant changes in transmission patterns. However, to truly understand the ecology of these populations, and how their ecological characteristics in livestock and the environment relate to transmission and virulence will require systematic sampling and accurate estimates of ST or cgMLST variant frequencies from those environments for robust comparison to those found in clinical samples. Epidemiological variants (i.e., cgMLST) making all the way to human clinical cases are by definition "successful". However, the cgMLST distribution, and pattern of dominance, could have arisen by random chance or founder effect, and subsequently be maintained by the influence of habitats that facilitate the survival and spread of a given variants (*Fraser, Hanage & Spratt, 2005*). Importantly, the ProkEvo platform should facilitate systematic collaboration and coupling of hierarchical-based genotypic analysis from ongoing surveillance studies and regulatory testing in animal and food production environments in order to generate actionable information.

In addition to bias in sample types, our analyses were also limited by the availability of standardized formats for associating metadata with WGS data in the SRA. Even the most basic type of information such as isolation date is not uniformly available or is not consistently entered into the same fields, which is required for automation. Ongoing efforts from consortia led by NIST and other agencies are making progress, but significant barriers to data sharing across regulatory, industry, and academic sectors still exist (*Sane & Edelstein, 2015*). However, we believe the opportunity for developing entirely new approaches to mitigation and control of pathogenic bacteria that will result from fundamental understanding of their ecology and evolution across entire food production systems will ultimately outweigh the risks of making such data and metadata publicly available. Indeed, experimental designs that incorporate systematic sampling and standardized metadata can then be coupled with modern statistical tools such as machine learning and pattern searching algorithms (*Wheeler, Gardner & Barquist, 2018*; *Schrider & Kern, 2018*; *Lupolova, Lycett & Gally, 2019*) that can be easily implemented in ProkEvo. Such approaches will enable communities of microbiologists, epidemiologists, and bacterial geneticists across the academic, regulatory, and industrial sectors to truly exploit the massive amount of emerging WGS data.

## CONCLUSIONS

In this paper we describe the **ProkEvo platform**, which is: **(1)** An automated, user-friendly, reproducible, and open-source platform for bacterial population genomics analyses that uses the Pegasus Workflow Management System; **(2)** A platform that can

scale the analysis from at least a few to tens of thousands of bacterial genomes using high-performance and high-throughput computational resources; **(3)** An easily modifiable and expandable platform that can accommodate additional steps, custom scripts and software, user databases, and species-specific data; **(4)** A modular platform that can run many thousands of analyses concurrently, if the resources are available; **(5)** A platform for which the memory and run time allocations are specified per job, and automatically increases its memory in the next retry; and **(6)** A pipeline that is distributed with conda environment and Docker image for all bioinformatics tools and databases needed to perform population genomics analyses in a reproducible fashion. Our case studies illustrate how to perform an initial, yet uniquely relevant for ecological and epidemiological inquiries, hierarchical-based population analyses using ProkEvo output files with reproducible Jupyter Notebooks and R scripts. Results from our case studies are clear illustrations of the types of population-based data mining that can be made from large-scale, WGS-based datasets using ProkEvo.

## ACKNOWLEDGEMENTS

This work was completed by utilizing the Holland Computing Center of the University of Nebraska, which receives support from the Nebraska Research Initiative, and using resources provided by the Open Science Grid, which is supported by the National Science Foundation and the U.S. Department of Energy's Office of Science. This research used the Pegasus Workflow Management Software funded by the National Science Foundation under grant #1664162. We would like to greatly thank Mats Rynge for his extensive assistance and valuable suggestions while setting up and running ProkEvo on the Open Science Grid. We also thank Dr. Derek Weitzel and Karan Vahi for their technical support.

This paper is dedicated to the memory of Dr. David Swanson, the former director of the Holland Computing Center, who passed away before this project was completed. Dr. David Swanson was an amazing individual, a true leader, and an inspiration to us all. He was a strong advocate of computational research and literacy, and passionate about finding ways to do large-scale science attainably, better, and faster. Our collaboration and shared endeavors would not have been possible without his support and guidance.

### Funding

This work was supported by funding from the IANR Agricultural Research Division and the National Institute for Antimicrobial Resistance Research and Education. The funders had no role in study design, data collection and analysis, decision to publish, or preparation of the manuscript.

### Grant Disclosures

The following grant information was disclosed by the authors:
IANR Agricultural Research Division.
National Institute for Antimicrobial Resistance Research and Education.

## Competing Interests

The authors declare that they have no competing interests.

## Author Contributions

- Natasha Pavlovikj conceived and designed the experiments, performed the experiments, analyzed the data, prepared figures and/or tables, authored or reviewed drafts of the paper, and approved the final draft.
- Joao Carlos Gomes-Neto conceived and designed the experiments, performed the experiments, analyzed the data, prepared figures and/or tables, authored or reviewed drafts of the paper, and approved the final draft.
- Jitender S. Deogun conceived and designed the experiments, authored or reviewed drafts of the paper, and approved the final draft.
- Andrew K. Benson conceived and designed the experiments, authored or reviewed drafts of the paper, and approved the final draft.

## Data Availability

The source code for ProkEvo is available at GitHub: https://github.com/npavlovikj/ProkEvo.

The Jupyter and R scripts used are available at GitHub: https://github.com/npavlovikj/ProkEvo/tree/master/jupyter_r_notebooks.

The input and intermediate files used for the analyses are available at figshare https://figshare.com/projects/ProkEvo/78612.

Links to the data are available in a Supplemental File.

## Supplemental Information

Supplemental information for this article can be found online at http://dx.doi.org/10.7717/peerj.11376#supplemental-information.

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
