# Peer review of "ProkEvo: an automated, reproducible, and scalable framework for high-throughput bacterial population genomics analyses"

_PeerJ, doi:10.7717/peerj.11376_

## Round 0.1 · original submission · Major Revisions

The reviewers have raised interest in this work; however, they have also raised major concerns particularly in a more comprehensive review of available literature and putting ProkEvo's novelty and innovation in their context. They have also commented on the availability of the method and its accessible installation and have specifically asked for more detail on parameter settings that could impact the utility of the pipeline.

Reviewer 1 ·

Basic reporting

The language used in this manuscript is clear, except that some sentences are too long to read. Also, all species names need to be italicised.

The authors failed to provide adequate background of the field. In particular, they ignored all the widely accepted core genome MLST schemes for Salmonella, Campylobacter and Staphylococcus, which are hosted in EnteroBase and pubmlst. They also falsely stated in line 560 that cgMLST is not scalable, ignoring the fact that both cgMLST platforms can easily handle genetic relationships of 1000's or 10,000's of genomes. The workaround proposed by the authors is, actually, not scalable, as they have to divide datasets into subsets of <2000 genomes each.

Experimental design

Overall speaking, ProkEvo is a well designed automated framework that helps people. The authors proven that, with the help of Pegasus and large clusters, ProkEvo can be used to handle 1,000's or 10,000's of genomes in a reasonable time.

I, however, have doubts on the reproducibility (or future-proof) of the framework. The authors assumed that all the programs implemented in ProkEov are eternal, and have not implemented (or described) a version system for these programs. Different versions of the same program can perform very differently. For example, the assemblies generated by SPAdes 3.6 is likely different from those by SPAdes 3.10. Without a careful, yet powerful version system in hand, scientists 5-year later will not be able to reproduce what they get today.

Secondly, ProkEvo is also not so "versatile". For example, ProkEvo by default accepts only assemblies with <=300 contigs and N50 >= 25Kb. However, if the same criteria are applied on Y. pestis or Shigella, many high quality genomes will fail because their chromosomes carry too many transposons. Authors will need to describe how these parameters can be adjusted. Do the users need to build up a new workflow with different settings, or can they be changed on the fly?

Finally, while most of the analyses in ProkEvo look like a cascading, automated workflow, the roary->fastbaps sub-pipeline requires some sort of human interactions. For example, the authors have manually selected subsets of <2000 genomes for this sub-pipeline. How does this interaction work in practice?

Validity of the findings

The authors described in detail about how the parallelization has improved the performance. However, it is still unclear to me that how long does every module in the workflow run in average and which program is the bottleneck. It will be great if the authors can list things out in a supplementary table.

It is also interesting that, the authors mentioned ABRicate as the program in ProkEov, but later referred the AMR results to "Resfinder". What exactly has been used?

The case studies are quite hard to follow, especially because the authors described populations using many different designations, but have not offered a figure to show the relationships between these population designations. There may be two trees in the middle of the rings in Figure 7? They, however, are not clear and large enough to show the branches. And why are there two figure legends for each of the ring?

Reviewer 2 ·

Basic reporting

In this manuscript, authors report the development of a framework for the large scale analysis of bacterial whole genome sequence data. The pipeline is split in two parts. The first part downloads raw sequence data from NCBI SRA and performs quality control of the data. The second part runs the actual population genomics analysis. The strength of the paper is that it appears to be designed work with very large datasets of over 20K genomes. However, there is limited innovation in the design. This seems to be another WGS pipeline that wraps several programs. They also dont compare this program with other recently published pipelines such as Bactopia. In fact these two looks very similar. Bactopia probably have more options such as the ability to run MASH.

Experimental design

From the methods section it is not clear how much flexibility this program has for input data. The default is to provide a list of SRA accession numbers as a file. Is there any option to load the data directly if the user has that in the local system? The other main problem with this paper is extremely poor documentation. methods section gives three different links to the GitHub pages for the different versions of the program. However, clicking on that takes to the same page. Additionally, the installation seems to be quite complicated with the need of manually specifying paths. Unless these are fixed, nobody other than the authors may be able to use the pipeline.

Validity of the findings

I could not find the conda ad docker packages in the Github page. So there is noway to run the pipeline to see whether it works

Additional comments

If the authors wish to see wide usage of their program, a much better documentation for the installation and use of the software needs to be provided. I spent quite a bit of time looking for the conda and docker versions mentioned in the paper. However, the links in the paper took me to the same page. The current documentation is poor and it was frustrating for me.

---

## Round 0.2 · accepted · Accept

Congratulations on the acceptance of your manuscript. In preparing the paper for publication, please confirm that the formatting and style match PeerJ's guidelines.

Reviewer 1 ·

Basic reporting

The revised manuscript describes ProkEvol, which is a useful automated framework for large scale genomic analysis. The authors have addressed most of the issues from previous review, except that they still have not italicized many of genus names. The authors listed https://peerj.com/about/author-instructions/#species-formatting, which describes: "The names of higher taxonomic levels (family, order, class, phylum or division, and kingdom) should be capitalized but not italicized. "

The genus names need to be italicized because they are not "higher taxonomic levels"

Experimental design

There is no remaining issue for the experimental design

Validity of the findings

The evaluations and examples are now well described.